# A cross-sectional and population-based study from primary care on post-COVID-19 conditions in non-hospitalized patients

Dominik J. Ose [1,2,5], Elena Gardner[1,5], Morgan Millar [3], Andrew Curtin[1], Jiqiang Wu[1], Mingyuan Zhang [4], Camie Schaefer[1], Jing Wang[1], Jennifer Leiser [1], Kirsten Stoesser [1] & Bernadette Kiraly [1✉]

**Abstract**

**Background** Current research on post-COVID-19 conditions (PCC) has focused on hospitalized COVID-19 patients, and often lacks a comparison group. This study assessed the prevalence of PCC in non-hospitalized COVID-19 primary care patients compared to primary care patients not diagnosed with COVID-19.

**Methods** This cross-sectional, population-based study ($n = 2539$) analyzed and compared the prevalence of PCC in patients with a positive COVID-19 test ($n = 1410$) and patients with a negative COVID-19 test ($n = 1129$) never hospitalized for COVID-19 related conditions. Participants were identified using electronic health records and completed an electronic questionnaire, available in English and Spanish, including 54 potential post COVID-19 symptoms. Logistic regression was conducted to assess the association of PCC with COVID-19.

**Results** Post-COVID-19 conditions are prevalent in both groups, and significantly more prevalent in patients with COVID-19. Strong significant differences exist for the twenty most reported conditions, except for anxiety. Common conditions are fatigue (59.5% (COVID-19 positive) vs. 41.3% (COVID-19 negative); OR 2.15 [1.79–2.60]), difficulty sleeping (52.1% (positive) vs. 41.9% (negative); OR 1.42 [1.18–1.71]) and concentration problems (50.6% (positive) vs 28.5% (negative); OR 2.64 [2.17–3.22]). Similar disparities in prevalence are also observed after comparing two groups (positive vs. negative) by age, sex, time since testing, and race/ethnicity.

**Conclusions** PCC is highly prevalent in non-hospitalized COVID-19 patients in primary care. However, it is important to note that PCC strongly overlaps with common health symptoms seen in primary care, including fatigue, difficulty sleeping, and headaches, which makes the diagnosis of PCC in primary care even more challenging.

**Plain Language Summary**

Research on post-COVID-19 conditions (PCC), also known as Long COVID, has often involved hospitalized COVID-19 patients. However, many patients with COVID-19 were not hospitalized, therefore how commonly the condition affects individuals attending primary care services is not accounted for. Here, we assessed non-hospitalized primary care patients with and without COVID-19. Our results demonstrate that PCC is highly common among primary care patients with COVID-19 and often presents as fatigue, difficulty sleeping, and concentration problems. As these symptoms overlap with other non-COVID-related conditions, it is challenging to accurately diagnose PCC. This calls for improved diagnostics and management of PCC in primary care settings, which is often the first point of contact with the healthcare systems for many patients.

[1] University of Utah Health, School of Medicine, Department of Family and Preventive Medicine, Salt Lake City, UT, USA. [2] Westsächsische Hochschule - Zwickau, Faculty of Health and Healthcare Science, Zwickau, Germany. [3] University of Utah Health, School of Medicine, Department of Internal Medicine, Salt Lake City, UT, USA. [4] University of Utah Health, Data Science Services, Salt Lake City, UT, USA. [5] These authors contributed equally: Dominik J. Ose, Elena Gardner. ✉email: bernadette.kiraly@hsc.utah.edu

Research about long-COVID, often referred to as post-COVID-19 conditions, is emerging[1]. Post-COVID-19 conditions (PCC) are characterized as signs and symptoms that develop during or after a COVID-19 infection, are consistently present for more than 12 weeks, and are not attributable to alternative diagnoses[2,3]. PCC can affect multiple body systems and affect various symptoms, including fatigue, shortness of breath, smell/taste disorders, muscle weakness, anxiety, and memory problems[4–8]. A meta-analysis indicated that 80% of patients with a COVID-19 infection developed one or more long-term symptoms[9].

Previously, many studies have focused on PCC in hospitalized patients and suggested that PCC is more common among, or even specific to, hospitalized COVID-19 patients[10–15]. On the other hand, research in non-hospitalized patients is evolving[7,16–18]; a current study suggests that hospitalized and non-hospitalized patients have similar rates of PCC[19]. However, both hospitalized and non-hospitalized patients with PCC often report poor or decreased quality of life regarding mobility, pain and discomfort, and the ability to return to normal levels of work or social activity[20–22]. Still, studies focusing exclusively on non-hospitalized or primary care patients are rare.

During the pandemic, primary care has been instrumental in identifying, managing, and monitoring patients with COVID-19 as well as has been critical for the implementation and mass delivery of vaccination[23,24]. Primary care, often the first point of contact with the health system, is also likely to play an important role in addressing challenges associated with PCC[25]. Several PCC-related symptoms (e.g., fatigue, muscle weakness, depression) are commonly reported and treated in primary care settings, independent of COVID-19[26].

However, evidence about PCC in primary care remains scarce. In particular, population-based studies with control groups that quantify the burden of PCC in primary care are missing. To address this situation, this study aims to analyze the prevalence of PCC in non-hospitalized COVID-19 patients in primary care, and to compare the prevalence of PCC symptoms between patients with and without COVID-19.

The results of our study show that PCC symptoms, such as fatigue, shortness of breath, and difficulty sleeping, are prevalent among non-hospitalized primary care patients, independent of COVID-19. However, the symptom burden is much higher among COVID-19 patients. This evidence highlights the major challenge faced by primary care providers – how to distinguish PCC from the background of symptoms commonly addressed in primary care. Overall, our findings support claims that PCC is ideally managed in the primary care setting, especially due to the holistic, longitudinal, and multidisciplinary aspects of primary care. In particular, comprehensive training on care pathways, guidelines, and referral criteria are necessary to support a primary care-led response to PCC.

## Methods

**Design and population**. This cross-sectional, population-based study was conducted at the University of Utah Health (U of U Health) system in Salt Lake City, Utah, United States. U of U Health is Utah's only academic healthcare system and provides primary care through 12 health centers in the greater Salt Lake City area. These clinics serve a combined total of about 120,000 patients annually. All participants have provided informed consent. The University of Utah Institutional Review Board (IRB #139714) exempted the study.

**Inclusion/exclusion criteria**. Participant selection criteria included: age 18+ years, at least one prior visit (in-person or virtual) with a U of U Health primary care center between January 2020 and March 2021, email address on file, preferred language English or Spanish, and a positive or negative COVID-19 test result (PCR) between March 1st, 2020, and August 31st, 2021, documented in their electronic health record (EHR). Patients were excluded if they had a COVID-19 test before March 1st, 2020, or if they were hospitalized or sought emergency department care related to COVID-19.

**Questionnaire**. The questionnaire was developed utilizing input from 1) a literature review to identify common post-COVID-19 symptoms and 2) primary care physician observations during the pandemic. The questionnaire development was iterative, with multiple drafts revised for clarity and content validity based on feedback from colleagues, clinicians, and other researchers with expertise in questionnaire methods. A pilot test of the questionnaire was conducted with faculty members at the Department of Family and Preventative Medicine at U of U Health to clarify and refine the contents and usability. The questionnaire was composed in English and translated into Spanish by a certified interpreter and native speaker. The questionnaire consisted of 54 PCC-related symptoms grouped into seven categories understandable to the public (Supplementary Table 1). Patients were asked to select symptoms they have experienced in the week prior and to rate the severity of those symptoms on a 3-point scale (mild, moderate, severe). Participants were not offered compensation for participating. Both the English and Spanish versions of the questionnaire started data collection on 08/31/2021 and ended 11/15/2021.

**Participants**. All clinics utilize a shared EHR system. Data from EHRs were stored in the University's Enterprise Data Warehouse (EDW). We identified 126,440 primary care patients in the EDW for possible inclusion in the study. Of those primary care patients, 124,606 were not hospitalized for COVID-19. We excluded patients from the non-hospitalized cohort with a preferred language other than English or Spanish ($n = 4084$). The remaining patients were split into patients who preferred English ($n = 114,588$) and patients who preferred Spanish ($n = 5934$). After excluding patients in both language groups with no COVID-19 test in the EHR, patients were further subdivided into English-preferred patients with a negative COVID-19 test ($n = 46,065$), English-preferred patients with a positive COVID-19 test ($n = 7356$), Spanish-preferred with a negative COVID-19 test ($n = 1700$), and Spanish-preferred with a positive COVID-19 test ($n = 905$). Participants tested for COVID-19 because they were experiencing symptoms within the last week. Among the remaining English-preferred cohort, 12,429 patients with a positive test and 7239 patients with a negative test were randomly selected to receive an invitation to complete the questionnaire (Supplementary Fig. 1). All of the patients in the Spanish-preferred cohort with test results received an invitation to complete the questionnaire. In total, 22,248 questionnaires were successfully delivered to patients by email sent through REDCap. 19,321 patient records did not respond to the questionnaire, and 2927 responses were submitted for the questionnaire (Fig. 1). Questionnaire records with duplicate responses ($n = 3$), and unfinished questionnaires ($n = 385$) were excluded from the final analysis. Finally, 2539 participants (1410 COVID-19 positive, 1129 COVID-19 negative) were verified and completed the survey responses for analysis of common post-COVID-19 symptoms.

## Measures

*Outcomes*. Common post-COVID-19 symptoms were reported by both COVID-19 positive and negative patients in their

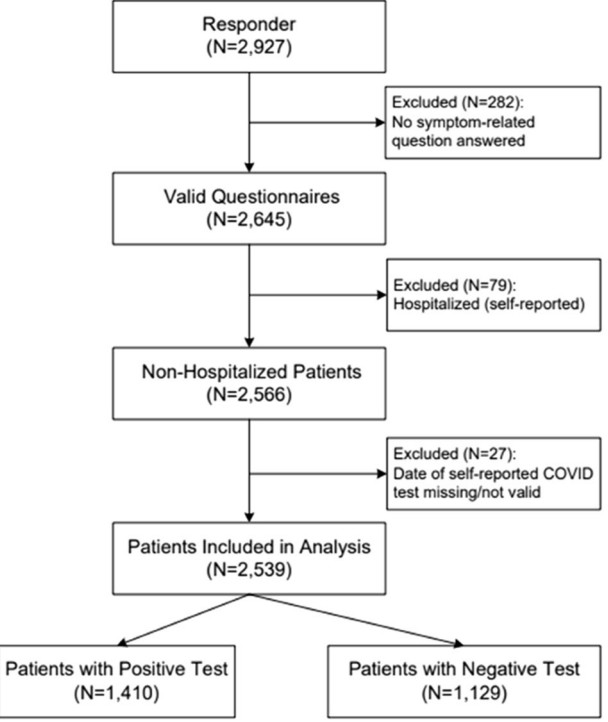

**Fig. 1** Flowchart of selecting patients with and without COVID-19 for analysis. Out of 2927 patients who responded, 2,539 patients were included in the analysis. Out of those, 1410 patients had a positive COVID-19 test and 1129 had a negative COVID-19 test respectively. Out of the 388 excluded patients, 282 were excluded for not answering any symptom-related question, 79 were excluded for a previous hospitalization, and 27 were excluded for a missing or invalid COVID-19 test date.

questionaries, classified into 7 categories, including (1) general symptoms (fatigue/tiredness, muscle & body aches, joint pains, shortness of breath, cough); (2) brain & nervous system headaches (concentration problems, memory problems, general weakness, dizziness, balance problems); (3) mental well-being (difficulty sleeping, anxiety, depression); (4) ears, nose, and throat (congested nose, ringing in ears); (5) heart or circulation (irregular heartbeats, leg pain when walking); (6) eyes or vision (dry eyes); (7) stomach or digestion (heartburn).

*Exposure.* The secondary COVID-19 test result (positive, negative) was documented in the EDW. In addition, participants could self-report a positive COVID-19 test result outside the University of Utah health system. If they did, the answers were included when defining COVID-19 test result.

*Covariates.* Information on the following secondary demographic and clinical characteristics was obtained from the EDW. *Sex* was defined as a biological variable to describe biological differences and influences when comparing males and females. Thus, *sex* was coded binary as male or female as opposed to gender, which could exist on a spectrum and is more often used when describing social or psychological differences between men, women, and other genders[27]. Other covariates included *age* (18–34 years, 35–49 years, 50 years and above), *race* (American Indian/Alaskan Native, Asian, Black/African American, Native Hawaiian/Other Pacific Islander, White, Other, Unknown), *ethnicity* (Hispanic/Latino or non-Hispanic/Latino), *BMI* (<18.50: underweight, 18.50–24.99: normal weight, 25.00–39.99: overweight, 40.00+: obese), *smoking status* (never smoked, quit smoking, currently smoking), *COVID-19 vaccine status* (none, any dose), *time*

between testing and questionnaire receipt (3–9 months, 10–12 months, more than 12 months), and *Charlson Comorbidity Index* (Scored 0–15).

**Power**. Given the sample of 2539 participants (1410 COVID-19 positive, 1129 COVID-19 negative) who reported common post-COVID-19 symptoms, power analysis was conducted to calculate power for the hypothesis test for a significant difference in the prevalence of PCC between COVID-19 positive and negative patients. Using a two-sided Chi-square test at a significance level of .05, we would have 93–99% power to detect a small effect size of 0.15–0.2.

**Statistical analysis**. Patients were categorized as having positive or negative COVID-19 tests. For the final analysis, symptom scores (0–3) were dichotomized into symptom not reported (0) and symptom reported (1–3). Descriptive statistics were calculated, including the frequency and percentage, for all categorical variables (demographic and clinical characteristics, PCC-related symptoms, COVID-19 test result, Charlson comorbidity index (CCI), and body mass index). The mean and standard deviation were calculated for continuous variables (age and CCI). The chi-square test was conducted to assess the associations of the COVID-19 test result with other variables (Table 1).

Logistic regression was conducted to assess the association of PCC with COVID-19, after the adjustment of covariates (age, sex, race, ethnicity, time since COVID-19 test and questionnaire receipt, smoking, COVID-19 vaccine status, and Charlson Comorbidity Index) (Table 2). Cross-validation was used to prevent overfitting, improve model performance, and address the potential limitation of self-selection bias. Model selection criteria and a comprehensive directed acyclic graph (DAG) of confounding variables were also considered in creating the logistic regression model. These analyses were then stratified by several clinicodemographic moderators, including age (Table 3), sex (Table 4), race (Table 5), ethnicity (Table 6), and time since COVID-19 test (Table 7) to assess the moderating effects of these variables on the association of PCC and COVID-19. Accordingly, the Tukey multiple comparison test was used to compare the prevalence between two groups by these moderators. The reported results included unadjusted and adjusted p values, odds ratios (OR), and associated 95% confidence intervals. The significance level was set to .05. All statistical tests were 2-sided. Missing values were removed using listwise deletion. All the above analyses were conducted in RStudio.

**Reporting summary**. Further information on research design is available in the Nature Portfolio Reporting Summary linked to this article.

## Results
**Clinical and demographic characteristics**. A total of 22,248 patients received the questionnaire, 2927 patients responded (13.2% response rate) and 2539 patients reported symptoms (Supplementary Fig. 1); 1410 with positive COVID-19 tests (55.5%) and 1129 with negative COVID-19 tests (44.5%). The mean age was 44.4 years, 63.3% of participants were female, 20.6% were Hispanic/Latino ($n = 523$), and 18.9% were non-white/non-Caucasian (Table 1). In contrast, non-responders were younger (43.0; $P < 0.001$), less often female (59.1%; $P < 0.001$) and less often non-white (37.7%; $P < 0.001$) (Supplementary Table 2). Compared to COVID-19 negative patients, COVID-19 positive patients were younger (42.6 years vs. 46.7 years; $P < 0.001$) and more likely to be males (37% vs. 35.7%; $P = 0.514$), non-

**Table 1 Secondary clinical and demographic characteristics**

| | Population | | Covid-19 test | | | | P-Value |
|---|---|---|---|---|---|---|---|
| | | | Positive | | Negative | | |
| | N | % | N | % | N | % | |
| Included in analysis | 2539 | 100 | 1410 | 55.5 | 1129 | 44.5 | |
| Age, mean (CI) | 44.4 | (13.8, 75) | 42.6 | (14,71.2) | 46.7 | (14.2,79.2) | <0.001 |
| *Age-Categorized* | | | | | | | <0.001 |
| 18–34 years | 810 | 31.9 | 476 | 33.8 | 334 | 29.6 | |
| 35–49 years | 797 | 31.4 | 486 | 34.5 | 311 | 27.5 | |
| 50 years and older | 932 | 36.7 | 448 | 31.8 | 484 | 42.9 | |
| *Sex* | | | | | | | 0.514 |
| Female | 1,615 | 63.6 | 889 | 63.0 | 726 | 64.3 | |
| Male | 924 | 36.4 | 521 | 37.0 | 403 | 35.7 | |
| *Ethnicity* | | | | | | | <0.001 |
| Non-Hispanic/Latino | 1,947 | 76.7 | 1,135 | 80.5 | 811 | 71.8 | |
| Hispanic/Latino | 523 | 20.6 | 234 | 16.6 | 289 | 25.6 | |
| Unknown | 69 | 2.7 | 41 | 2.9 | 29 | 2.6 | |
| *Race* | | | | | | | 0.032 |
| White/Caucasian | 2,060 | 81.1 | 1,165 | 82.6 | 895 | 79.3 | |
| Non-White/Non-Caucasian[a] | 479 | 18.9 | 245 | 17.4 | 234 | 20.7 | |
| CCI[b], mean (CI) | 1.01 | (0.95, 1.07) | 0.91 | (0.83,0.99) | 1.14 | (1.04, 1.24) | <0.001 |
| *Smoking Status[c]* | | | | | | | 0.576 |
| Never | 2,021 | 79.6 | 1,125 | 80.6 | 896 | 80.0 | |
| Quit | 404 | 15.9 | 222 | 15.9 | 182 | 16.3 | |
| Yes | 89 | 3.5 | 47 | 3.4 | 42 | 3.8 | |
| Unknown | 25 | 0.0 | 2 | 0.1 | 0 | 0.0 | |
| BMI, mean (CI) | 29.6 | (13.7,45.5) | 29.9 | (15, 44.8) | 29.2 | (12.3, 46) | |
| *BMI (Kg/m²)* | | | | | | | <0.001 |
| Underweight (<18.50) | 28 | 1.1 | 15 | 1.1 | 13 | 1.2 | |
| Normal Weight (18.50-24.99) | 678 | 26.7 | 344 | 24.4 | 334 | 29.6 | |
| Overweight (25.00-39.99) | 1,356 | 53.4 | 759 | 53.8 | 597 | 52.9 | |
| Obese (40.00+) | 229 | 9.0 | 133 | 9.4 | 96 | 8.5 | |
| Unknown | 248 | 9.8 | 159 | 11.3 | 89 | 7.9 | |
| *Vaccination status[d]* | | | | | | | 0.001 |
| None | 224 | 8.8 | 166 | 11.8 | 58 | 5.1 | |
| Yes, Any | 2,315 | 91.2 | 1,244 | 88.2 | 1,071 | 94.9 | |
| *Time After COVID-19 Test* | | | | | | | <0.001 |
| 3-9 months | 765 | 30.1 | 427 | 30.3 | 338 | 29.9 | |
| 10-12 months | 1,073 | 42.3 | 698 | 49.5 | 375 | 33.2 | |
| Over a Year | 701 | 27.6 | 285 | 20.2 | 416 | 36.8 | |

[a]Non-white/Non-Caucasian: American Indian/Alaskan Native (n = 17), Asian (n = 46), Black/African American (n = 27), Native Hawaiian/Other Pacific Islander (n = 45), Other race (n = 344)
[b]Charlson Comorbidity Index, there were missing values among survey responses for CCI (n = 4) with 2 in the COVID-positive group and 2 in the COVID=negative group. Patient CCI Scores range 0–15, IQR: 1. Age range 18–91. BMI range 13–74. The standard error of the mean was calculated to test the mean of the sampling distribution and for the calculation of the 95% confidence interval (CI).
[c]Smoking status reported as "passive" treated as "yes". Smoking status had n = 16 missing values in the COVID-positive group and n = 9 missing values in the COVID-negative group.
[d]Vaccination status "yes, any" is any record or self-report of a single dose or more of any available vaccine J&J, Pfizer, Moderna, etc. BMI had n = 159 missing values in the COVID-positive group and n = 89 missing values in the COVID-negative group.

Hispanics/Latinos (80.5% vs. 71.8%; P < 0.001), and unvaccinated persons (11.8% vs. 5.1%; P < 0.001) (Table 1).

**Common post COVID-19 conditions**. There were 20 common symptoms reported by both COVID-19-positive and negative patients. Among them, the following symptoms had relatively higher prevalence than the others, including fatigue/tiredness (51.4%), anxiety (48.3%), difficulty sleeping (47.6%), headaches (44.1%), and concentration problems (40.8%). Balance problems (18.9%) and leg pain when walking (18.8%) had the lowest prevalence. All symptoms were more common among patients with positive COVID-19 tests. For example, the prevalence of common symptoms and odds ratios (OR) comparing COVID-19 positive and negative patients were, respectively, concentration problems (50.6% vs. 28.5%; 2.64 [2.17–3.22]), memory problems (39.4% vs. 19.2%; 2.65 [2.15–3.28], fatigue/tiredness (59.5% vs. 41.3%; 2.15 [1.79–2.60]), headaches (49.8.% vs. 37%, 1.60 [1.32–1.94]),

difficulty sleeping (52.1% vs. 41.9%, 1.42 [1.18–1.71]), and anxiety (51.1% vs. 44.7%, 1.18 [0.98–1.42]) (Table 2, Fig. 2).

These odds ratios were also presented graphically in Supplementary Fig. 2. Differences between COVID-19 positive and COVID-19 negative groups are more clearly shown in this figure. For example, group differences for shortness of breath, concentration problems, memory problems were similar and higher than group differences for other symptoms. Group differences for mental well-being symptoms (difficulty sleeping, anxiety, depression), dry eyes, and heartburn (reflux) were similar and lower than group differences for other symptoms. Group differences for muscle & body aches, joint pains, general weakness, dizziness balance problems, irregular heartbeats, and leg pain when walking were also similar.

Figure 3 (Supplementary Table 3) shows descending absolute differences in prevalence of the top 20 most frequently experienced symptoms between COVID-19 positive and COVID-19 negative patients. Differences for concentration problems, memory problems,

**Table 2 Most common post COVID-19 symptoms (n = 2539)**

| Symptom | Population | | | By COVID-19 test result | | | | | | P-value | | Odds ratio |
|---|---|---|---|---|---|---|---|---|---|---|---|---|
| | | | | Positive | | | Negative | | | Unadjusted | Adjusted[c] | [95% CI] |
| | N[a] | Yes[b] | % | N[a] | Yes[b] | % | N[a] | Yes[b] | % | | | |
| *General symptoms* | | | | | | | | | | | | |
| Fatigue/Tiredness | 2281 | 1173 | 51.4 | 1267 | 754 | 59.5 | 1,014 | 419 | 41.3 | <0.001 | <0.001 | 2.15 [1.79–2.60] |
| Muscle & body aches | 2270 | 724 | 31.9 | 1260 | 471 | 37.4 | 1,010 | 253 | 25.0 | <0.001 | <0.001 | 1.83 [1.49–2.24] |
| Joint pains | 2269 | 700 | 30.9 | 1261 | 456 | 36.2 | 1,008 | 244 | 24.2 | <0.001 | <0.001 | 1.99 [1.62–2.46] |
| Shortness of breath | 2272 | 527 | 23.2 | 1261 | 384 | 30.5 | 1,011 | 143 | 14.1 | <0.001 | <0.001 | 2.68 [2.13–3.39] |
| Cough | 2263 | 525 | 23.2 | 1253 | 330 | 26.3 | 1,010 | 195 | 19.3 | <0.001 | <0.001 | 1.55 [1.25–1.94] |
| *Brain & nervous system* | | | | | | | | | | | | |
| Headaches | 2215 | 977 | 44.1 | 1224 | 610 | 49.8 | 991 | 367 | 37.0 | <0.001 | <0.001 | 1.60 [1.32–1.94] |
| Concentration problems | 2228 | 908 | 40.8 | 1236 | 625 | 50.6 | 992 | 283 | 28.5 | <0.001 | <0.001 | 2.64 [2.17–3.22] |
| Memory problems | 2226 | 677 | 30.4 | 1233 | 486 | 39.4 | 993 | 191 | 19.2 | <0.001 | <0.001 | 2.65 [2.15–3.28] |
| General weakness | 2224 | 487 | 21.9 | 1233 | 332 | 26.9 | 991 | 155 | 15.6 | <0.001 | <0.001 | 2.03 [1.61–2.57] |
| Dizziness | 2219 | 452 | 20.4 | 1229 | 305 | 24.8 | 990 | 147 | 14.8 | <0.001 | <0.001 | 1.92 [1.52–2.43] |
| Balance problems | 2222 | 419 | 18.9 | 1231 | 281 | 22.8 | 991 | 138 | 13.9 | <0.001 | <0.001 | 2.07 [1.62–2.66] |
| *Mental well-being* | | | | | | | | | | | | |
| Difficulty sleeping | 2216 | 1,054 | 47.6 | 1234 | 643 | 52.1 | 982 | 411 | 41.9 | <0.001 | <0.001 | 1.42 [1.18–1.71] |
| Anxiety | 2213 | 1,068 | 48.3 | 1229 | 628 | 51.1 | 984 | 440 | 44.7 | <0.001 | 0.086 | 1.18 [0.98–1.42] |
| Depression | 2219 | 775 | 34.9 | 1233 | 467 | 37.9 | 986 | 308 | 31.2 | 0.001 | 0.040 | 1.22 [1.01–1.48] |
| *Ears, nose, and throat* | | | | | | | | | | | | |
| Congested nose | 2260 | 775 | 34.3 | 1251 | 468 | 37.4 | 1,009 | 307 | 30.4 | <0.001 | 0.008 | 1.28 [1.07–1.54] |
| Ringing in ears | 2250 | 494 | 22.0 | 1245 | 303 | 24.3 | 1,005 | 191 | 19.0 | 0.002 | 0.001 | 1.42 [1.15–1.76] |
| *Heart or circulation* | | | | | | | | | | | | |
| Irregular heartbeats | 2242 | 447 | 19.9 | 1243 | 311 | 25.0 | 999 | 136 | 13.6 | <0.001 | <0.001 | 2.19 [1.73–2.80] |
| Leg pain when walking | 2239 | 421 | 18.8 | 1241 | 287 | 23.1 | 998 | 134 | 13.4 | <0.001 | <0.001 | 2.08 [1.62–2.67] |
| *Eyes or vision* | | | | | | | | | | | | |
| Dry eyes | 2238 | 585 | 26.1 | 1242 | 349 | 28.1 | 996 | 236 | 23.7 | 0.019 | <0.001 | 1.44 [1.17–1.78] |
| *Stomach or digestion* | | | | | | | | | | | | |
| Heartburn (reflux) | 2238 | 534 | 23.9 | 1237 | 334 | 27.0 | 1,001 | 200 | 20.0 | <0.001 | <0.001 | 1.47 [1.20–1.81] |

[a]Responder (Reported yes symptom or no symptom).
[b]N Reported patient with symptom.
[c]P-Value calculation by Logistic Regression adjusted for age, sex, BMI, vaccine status, race/ethnicity, CCI, and time after COVID-19 test, respectively. See Supplementary Fig. 2 for a graphical representation of odds ratio and 95% confidence interval.

**Table 3 Frequently reported symptoms by age group (n = 2539)**

| Symptom | 18–34 years (n = 810) | | | | | 35–49 years (n = 797) | | | | | 50 years and above (n = 932) | | | | |
|---|---|---|---|---|---|---|---|---|---|---|---|---|---|---|---|
| | COVID-19 test result | | | | | COVID-19 test result | | | | | COVID-19 test result | | | | |
| | Positive | | Negative | | Odds Ratio | Positive | | Negative | | Odds ratio | Positive | | Negative | | Odds ratio |
| | Nª | % | Nª | % | [95% CI] | Nª | % | Nª | % | [95% CI] | Nª | % | Nª | % | [95% CI] |
| *General symptoms* | | | | | | | | | | | | | | | |
| Fatigue/Tiredness | 283 | 59.7 | 169 | 50.6 | 1.64 [1.19–2.27] | 305 | 63.3 | 133 | 43.3 | 2.26 [1.64–3.13] | 240 | 54.4 | 159 | 33.2 | 2.51 [1.86–3.38] |
| Muscle & body aches | 153 | 32.3 | 91 | 27.2 | 1.28 [0.91–1.81] | 199 | 41.5 | 79 | 25.7 | 2.17 [1.53–3.09] | 161 | 36.7 | 110 | 23.1 | 2.00 [1.45–2.77] |
| Joint pains | 136 | 28.7 | 59 | 17.8 | 1.89 [1.30–2.78] | 192 | 39.8 | 75 | 24.4 | 2.26 [1.59–3.25] | 169 | 38.9 | 129 | 27.1 | 1.91 [1.39–2.61] |
| Shortness of breath | 134 | 28.3 | 52 | 15.7 | 1.98 [1.58–2.92] | 168 | 35.1 | 43 | 14.0 | 3.64 [2.42–5.56] | 126 | 28.6 | 63 | 13.2 | 2.98 [2.06–4.36] |
| Cough | 126 | 26.6 | 70 | 21.1 | 1.28 [0.90–1.84] | 122 | 25.6 | 54 | 17.5 | 1.68 [1.13–2.53] | 109 | 25.0 | 91 | 19.1 | 1.61 [1.14–2.29] |
| *Brain & nervous system* | | | | | | | | | | | | | | | |
| Headaches | 246 | 54.2 | 137 | 42.9 | 1.59 [1.15–2.20] | 264 | 56.7 | 146 | 47.7 | 1.42 [1.02–1.96] | 163 | 37.7 | 121 | 25.9 | 1.60 [1.17–2.20] |
| Concentration problems | 235 | 51.8 | 118 | 37.0 | 2.13 [1.54–2.95] | 264 | 56.1 | 94 | 39.2 | 2.90 [2.07–4.08] | 189 | 43.1 | 99 | 21.1 | 2.78 [2.02–3.85] |
| Memory problems | 157 | 34.5 | 69 | 21.6 | 1.80 [1.26–2.59] | 203 | 43.5 | 59 | 19.3 | 3.03 [2.30–4.82] | 174 | 39.6 | 87 | 18.5 | 2.64 [1.91–3.67] |
| General weakness | 105 | 23.0 | 45 | 14.2 | 1.68 [1.12–2.54] | 139 | 29.7 | 56 | 18.3 | 2.14 [1.45–3.21] | 125 | 28.6 | 69 | 14.7 | 2.36 [1.65–3.41] |
| Dizziness | 105 | 23.0 | 49 | 15.5 | 1.69 [1.14–2.54] | 129 | 27.7 | 54 | 17.6 | 1.80 [1.22–2.70] | 98 | 22.5 | 64 | 13.6 | 1.71 [1.18–2.50] |
| Balance problems | 75 | 16.5 | 32 | 10.1 | 1.72 [1.08–2.79] | 107 | 23.1 | 36 | 11.8 | 2.30 [1.47–3.66] | 122 | 27.8 | 89 | 19.0 | 1.87 [1.32–2.66] |
| *Mental well-being* | | | | | | | | | | | | | | | |
| Difficulty sleeping | 248 | 54.5 | 161 | 50.6 | 1.06 [0.77–1.46] | 256 | 54.8 | 126 | 41.9 | 1.65 [1.19–2.29] | 201 | 45.8 | 169 | 36.4 | 1.49 [1.11–1.99] |
| Anxiety | 279 | 61.3 | 198 | 62.1 | 0.87 [0.62–1.21] | 266 | 57.0 | 148 | 49.2 | 1.34 [0.97–1.86] | 146 | 33.6 | 128 | 27.6 | 1.31 [0.95–1.81] |
| Depression | 209 | 45.8 | 148 | 46.4 | 0.89 [0.64–1.23] | 193 | 41.3 | 99 | 32.9 | 1.45 [1.04–2.05] | 112 | 25.7 | 91 | 19.5 | 1.46 [1.02–2.07] |
| *Ears, nose, and throat* | | | | | | | | | | | | | | | |
| Congested nose | 181 | 38.9 | 114 | 34.4 | 1.15 [0.84–1.59] | 185 | 38.7 | 97 | 31.5 | 1.36 [0.98–1.90] | 145 | 33.1 | 136 | 28.6 | 1.31 [0.96–1.79] |
| Ringing in ears | 109 | 23.6 | 56 | 17.0 | 1.50 [1.02–2.23] | 112 | 23.5 | 54 | 17.5 | 1.55 [1.04–2.35] | 118 | 27.0 | 104 | 22.1 | 1.27 [0.92–1.77] |
| *Heart or circulation* | | | | | | | | | | | | | | | |
| Irregular heartbeats | 107 | 23.3 | 57 | 17.6 | 1.41 [0.96–2.09] | 141 | 29.9 | 46 | 15.0 | 2.84 [1.90–4.30] | 90 | 20.5 | 50 | 10.6 | 2.24 [1.49–3.39] |
| Leg pain when walking | 89 | 19.3 | 41 | 12.7 | 1.47 [0.95–2.29] | 117 | 24.9 | 39 | 12.7 | 2.72 [1.75–4.33] | 104 | 23.7 | 65 | 13.8 | 2.14 [1.47–3.15] |
| *Eyes or vision* | | | | | | | | | | | | | | | |
| Dry eyes | 108 | 34.3 | 60 | 18.6 | 1.44 [0.98–2.12] | 132 | 28.3 | 63 | 20.7 | 1.65 [1.13–2.43] | 142 | 32.4 | 138 | 29.2 | 1.25 [0.91–1.70] |
| *Stomach or digestion* | | | | | | | | | | | | | | | |
| Heartburn (reflux) | 125 | 27.3 | 61 | 18.9 | 1.65 [1.14–2.42] | 138 | 29.6 | 68 | 22.1 | 1.50 [1.03–2.18] | 108 | 24.5 | 91 | 19.0 | 1.26 [0.89–1.79] |

ªResponder (Reported yes symptom); Odds ratio and 95% confidence interval by Logistic Regression adjusted for age, sex, BMI, vaccine status, race/ethnicity, CCI, and time after COVID-19 test respectively. 18–34 years as reference category.

**Table 4 Frequently reported symptoms by sex (n = 2539)**

| Symptom | Female (n = 1615) | | | | | Male (n = 924) | | | | | Comparison[a] | |
|---|---|---|---|---|---|---|---|---|---|---|---|---|
| | All | By COVID-19 test result | | | | All | By COVID-19 test result | | | | P-value | |
| | | Positive | | Negative | | | Positive | | Negative | | | |
| | % | N[b] | % | N[b] | % | Odds Ratio [95% CI] | % | N[b] | % | N[b] | % | Odds Ratio [95% CI] | Unadjusted | Adjusted[c] |

| Symptom | % | N[b] (Pos) | % (Pos) | N[b] (Neg) | % (Neg) | Odds Ratio [95% CI] | % (All) | N[b] (Pos) | % (Pos) | N[b] (Neg) | % (Neg) | Odds Ratio [95% CI] | Unadjusted | Adjusted[c] |
|---|---|---|---|---|---|---|---|---|---|---|---|---|---|---|
| *General symptoms* | | | | | | | | | | | | | | |
| Fatigue/tiredness | 56.4 | 576 | 65.5 | 335 | 46.5 | 2.16 [1.73–2.69] | 40.9 | 252 | 48.7 | 126 | 31.5 | 2.07 [1.53–2.81] | 0.000 | <0.001 |
| Muscle & body aches | 34.1 | 360 | 40.9 | 191 | 26.5 | 1.80 [1.42–2.28] | 26.2 | 153 | 29.9 | 89 | 22.5 | 1.74 [1.25–2.46] | 0.001 | <0.001 |
| Joint pains | 33.1 | 353 | 40.1 | 181 | 25.2 | 2.17 [1.71–2.76] | 24.5 | 144 | 28.1 | 82 | 20.7 | 1.71 [1.21–2.43] | <0.001 | <0.001 |
| Shortness of breath | 24.1 | 288 | 32.7 | 101 | 14.0 | 3.09 [2.36–4.09] | 21.3 | 140 | 27.4 | 57 | 14.4 | 2.14 [1.48–3.13] | 0.234 | 0.253 |
| Cough | 23.6 | 244 | 27.8 | 137 | 19.0 | 1.62 [1.25–2.10] | 20.7 | 113 | 22.2 | 78 | 19.7 | 1.32 [0.92–1.90] | 0.152 | 0.230 |
| *Brain & nervous system* | | | | | | | | | | | | | | |
| Headaches | 49.2 | 495 | 57.7 | 300 | 42.9 | 1.61 [1.29–2.01] | 30.5 | 178 | 36 | 104 | 26.5 | 1.52 [1.10–2.10] | <0.001 | <0.001 |
| Concentration problems | 44.8 | 487 | 56.2 | 236 | 33.7 | 2.43 [1.94–3.05] | 29.9 | 201 | 40.4 | 75 | 19.1 | 3.08 [2.20–4.37] | <0.001 | <0.001 |
| Memory problems | 32.3 | 373 | 43.1 | 149 | 21.3 | 2.60 [2.04–3.32] | 24.6 | 161 | 32.5 | 66 | 16.8 | 2.49 [1.75–3.57] | 0.002 | <0.001 |
| General weakness | 21.8 | 243 | 28.1 | 109 | 15.6 | 2.17 [1.65–2.86] | 20.2 | 126 | 25.4 | 61 | 15.5 | 1.87 [1.29–2.75] | 0.762 | 0.399 |
| Dizziness | 22.7 | 249 | 29.0 | 118 | 16.8 | 1.90 [1.46–2.47] | 14.3 | 83 | 16.7 | 49 | 12.5 | 1.47 [0.98–2.24] | <0.001 | <0.001 |
| Balance problems | 19.9 | 214 | 24.9 | 107 | 15.3 | 1.94 [1.46–2.59] | 15.2 | 90 | 18.1 | 50 | 12.7 | 1.76 [1.16–2.68] | 0.039 | <0.001 |
| *Mental well-being* | | | | | | | | | | | | | | |
| Difficulty sleeping | 49.6 | 478 | 55.5 | 323 | 46.5 | 1.26 [1.01–1.57] | 39.0 | 227 | 45.4 | 133 | 34.2 | 1.62 [1.20–2.20] | 0.005 | <0.001 |
| Anxiety | 52.6 | 497 | 57.7 | 353 | 50.9 | 1.12 [0.89–1.40] | 34.1 | 194 | 39.3 | 121 | 31.1 | 1.32 [0.96–1.82] | <0.001 | <0.001 |
| Depression | 38.3 | 367 | 42.7 | 252 | 36.2 | 1.18 [0.93–1.48] | 25.2 | 147 | 29.5 | 86 | 22.0 | 0.99 [0.94–1.90] | <0.001 | <0.001 |
| *Ears, nose, and throat* | | | | | | | | | | | | | | |
| Congested nose | 35.5 | 352 | 40.1 | 222 | 31.0 | 1.46 [1.17–1.82] | 30.7 | 159 | 31.5 | 125 | 31.3 | 0.98 [0.72–1.35] | 0.017 | 0.018 |
| Ringing in ears | 21.2 | 208 | 23.9 | 134 | 18.8 | 1.41 [1.09–1.84] | 22.8 | 131 | 25.9 | 80 | 20.3 | 1.41 [1.00–2.00] | 0.876 | 0.438 |
| *Heart or circulation* | | | | | | | | | | | | | | |
| Irregular heartbeats | 21.9 | 247 | 28.4 | 107 | 15.1 | 2.30 [1.76–3.02] | 14.8 | 91 | 18.1 | 46 | 11.7 | 1.60 [1.05–2.46] | <0.001 | <0.001 |
| Leg pain when walking | 18.8 | 219 | 25.3 | 85 | 12.0 | 2.58 [1.91–3.50] | 16.3 | 91 | 18.1 | 60 | 15.3 | 1.34 [0.90–2.00] | 0.024 | 0.371 |
| *Eyes or vision* | | | | | | | | | | | | | | |
| Dry eyes | 29.5 | 279 | 32.1 | 197 | 27.9 | 1.31 [1.03–1.66] | 18.1 | 103 | 20.6 | 64 | 16.2 | 1.65 [1.13–2.41] | <0.001 | <0.001 |
| *Stomach or digestion* | | | | | | | | | | | | | | |
| Heartburn (reflux) | 25.6 | 258 | 29.8 | 155 | 21.9 | 1.47 [1.15–1.89] | 19.2 | 113 | 22.7 | 64 | 16.2 | 1.47 [1.02–2.14] | 0.045 | 0.013 |

aFemale with a positive COVID-19 test result versus male with a positive Covid-19 test result.
bN Reported patient with symptom.
cAdjusted P-Value calculation by Logistic Regression adjusted for age, sex, BMI, vaccine status, race/ethnicity, CCI, and time after COVID-19 test, respectively.

**Table 5 Frequently reported symptoms by race (n = 2539)**

| Symptom | White/Caucasian (n = 2060) | | | | | | Non-White/Non-Caucasian (n = 479)[a] | | | | | | Comparison[b] | |
| | All | By COVID-19 test result | | | | Odds Ratio | All | By COVID-19 test result | | | | Odds Ratio | P-value | P-value |
| | | Positive | | Negative | | | | Positive | | Negative | | | | |
| | % | N[c] | % | N[c] | % | [95% CI] | % | N[c] | % | N[c] | % | [95% CI] | Unadjusted | Adjusted[d] |
| *General symptoms* | | | | | | | | | | | | | | |
| Fatigue/Tiredness | 51.4 | 681 | 58.8 | 377 | 42.4 | 1.96 [1.61-2.38] | 48.0 | 147 | 61.5 | 83 | 36.2 | 3.12 [2.05-4.80] | 0.897 | 0.640 |
| Muscle & body aches | 30.3 | 401 | 34.8 | 224 | 25.2 | 1.59 [1.29-1.98] | 35.1 | 112 | 46.5 | 56 | 24.6 | 2.84 [1.83-4.47] | 0.008 | 0.642 |
| Joint pains | 30.2 | 406 | 35.3 | 217 | 24.5 | 1.85 [1.49-2.30] | 28.6 | 91 | 37.8 | 46 | 20.2 | 2.99 [1.88-4.85] | 0.910 | 0.756 |
| Shortness of breath | 23.2 | 347 | 30.1 | 130 | 14.7 | 2.50 [1.96-3.21] | 22.8 | 81 | 33.9 | 28 | 12.2 | 3.98 [2.40-6.77] | 0.719 | 0.677 |
| Cough | 22.0 | 283 | 24.7 | 171 | 19.3 | 1.40 [1.11-1.77] | 24.4 | 74 | 30.8 | 43 | 18.9 | 2.11 [1.32-3.41] | 0.276 | 0.405 |
| *Brain & nervous system* | | | | | | | | | | | | | | |
| Headaches | 42.3 | 550 | 49.0 | 321 | 36.7 | 1.49 [1.21-1.82] | 42.8 | 123 | 53.7 | 82 | 37.6 | 1.97 [1.29-3.05] | 0.641 | 0.148 |
| Concentration problems | 39.6 | 565 | 49.9 | 250 | 28.5 | 2.49 [2.03-3.07] | 38.2 | 123 | 53.2 | 60 | 27.6 | 3.13 [2.02-4.90] | 0.841 | 0.951 |
| Memory problems | 29.4 | 435 | 38.5 | 170 | 19.4 | 2.48 [1.99-3.10] | 30.1 | 99 | 42.9 | 45 | 20.7 | 2.83 [1.80-4.51] | 0.669 | 0.888 |
| General weakness | 20.6 | 292 | 25.8 | 132 | 15.1 | 1.96 [1.53-2.52] | 24.0 | 77 | 33.5 | 38 | 17.5 | 2.51 [1.55-4.13] | 0.125 | 0.972 |
| Dizziness | 19.4 | 266 | 23.6 | 134 | 15.3 | 1.64 [1.28-2.11] | 20.7 | 66 | 28.7 | 33 | 15.1 | 2.41 [1.44-4.10] | 0.442 | 0.200 |
| Balance problems | 18.3 | 250 | 22.2 | 128 | 14.6 | 1.78 [1.38-2.31] | 17.3 | 54 | 23.5 | 29 | 13.5 | 2.46 [1.43-4.34] | 0.980 | 0.709 |
| *Mental well-being* | | | | | | | | | | | | | | |
| Difficulty sleeping | 46.1 | 578 | 51.2 | 371 | 42.7 | 1.29 [1.06-1.57] | 44.3 | 127 | 54.7 | 85 | 39.7 | 1.74 [1.15-2.66] | 0.815 | 0.978 |
| Anxiety | 46.3 | 568 | 48.8 | 385 | 44.3 | 1.11 [0.90-1.36] | 44.1 | 123 | 53.5 | 88 | 41.3 | 1.43 [0.92-2.21] | 0.638 | 0.086 |
| Depression | 33.8 | 423 | 37.5 | 274 | 31.5 | 1.19 [0.96-1.48] | 32.4 | 91 | 39.6 | 64 | 29.6 | 1.28 [0.82-2.01] | 0.949 | 0.840 |
| *Ears, nose, and throat* | | | | | | | | | | | | | | |
| Congested nose | 34.3 | 417 | 36.5 | 289 | 32.6 | 1.16 [0.95-1.42] | 31.5 | 94 | 39.7 | 57 | 25.1 | 1.99 [1.29-3.10] | 0.835 | 0.724 |
| Ringing in ears | 22.1 | 279 | 24.5 | 177 | 20.1 | 1.33 [1.06-1.68] | 20.0 | 60 | 25.4 | 36 | 15.9 | 1.90 [1.15-3.18] | 0.994 | 0.706 |
| *Heart or circulation* | | | | | | | | | | | | | | |
| Irregular heartbeats | 19.3 | 270 | 23.7 | 127 | 14.4 | 1.85 [1.44-2.38] | 19.4 | 68 | 29.1 | 25 | 11.3 | 3.57 [2.10-6.27] | 0.384 | 0.912 |
| Leg pain when walking | 16.5 | 232 | 20.4 | 108 | 12.3 | 1.94 [1.48-2.56] | 24.0 | 78 | 33.3 | 37 | 17.0 | 2.40 [1.48-3.95] | <0.001 | 0.026 |
| *Eyes or vision* | | | | | | | | | | | | | | |
| Dry eyes | 25.9 | 310 | 27.3 | 224 | 25.5 | 1.25 [1.00-1.56] | 22.8 | 72 | 30.6 | 37 | 16.7 | 2.65 [1.62-4.38] | 0.788 | 0.118 |
| *Stomach or digestion* | | | | | | | | | | | | | | |
| Heartburn (reflux) | 23.7 | 308 | 27.2 | 180 | 20.5 | 1.35 [1.08-1.70] | 21.3 | 63 | 27.3 | 39 | 17.5 | 2.13 [1.29-3.59] | 1.000 | 0.821 |

[a]Non-white/Non-Caucasian categorized as American Indian/Alaskan Native (n = 17), Asian (n = 46), Black/African American (n = 27), Native Hawaiian/Other Pacific Islander (n = 45), Other (n = 344).
[b]White/Caucasian with a positive COVID-19 test result versus Non-White/Non-Caucasian with a positive Covid-19 test result.
[c]N Reported patient with symptoms.
[d]Adjusted P-Value for comparing white and non-white COVID-positive patients calculated from Logistic Regression after the adjustment of age, sex, BMI, vaccine status, race/ethnicity, CCI, and time after COVID-19 test.

**Table 6 Frequently reported symptoms by ethnicity (n = 2539)**

| Symptoms | Hispanic/Latino (n = 523) | | | | | Non-Hispanic/Latino (n = 1947) | | | | | Comparison[a] | |
| | All | By COVID-19 test result | | | | All | By COVID-19 test result | | | | P-value | |
| | | Positive | | Negative | | | Positive | | Negative | | | |
| | % | N[b] | % | N[b] | % | Odds Ratio [95% CI] | % | N[b] | % | N[b] | % | Odds Ratio [95% CI] | Unadjusted | Adjusted[c] |
|---|---|---|---|---|---|---|---|---|---|---|---|---|---|---|
| *General symptoms* | | | | | | | | | | | | | | |
| Fatigue/Tiredness | 50.3 | 144 | 62.1 | 119 | 41.6 | 2.28 [1.53-3.40] | 51.2 | 667 | 59.4 | 329 | 40.8 | 2.12 [1.73-2.60] | 0.900 | 0.619 |
| Muscle & body aches | 33.7 | 107 | 46.5 | 69 | 24.1 | 2.99 [1.97-4.59] | 30.6 | 394 | 35.2 | 201 | 25.0 | 1.62 [1.30-2.02] | 0.015 | 0.131 |
| Joint pains | 26.6 | 81 | 35.1 | 58 | 20.4 | 2.45 [1.57-3.85] | 30.7 | 402 | 35.9 | 196 | 24.4 | 1.91 [1.52-2.39] | 0.997 | 0.452 |
| Shortness of breath | 22.0 | 75 | 32.8 | 40 | 13.9 | 3.31 [2.06-5.39] | 23.3 | 341 | 30.4 | 113 | 14.1 | 2.60 [2.02-3.37] | 0.916 | 0.511 |
| Cough | 23.9 | 64 | 28.1 | 61 | 21.4 | 1.48 [0.94-2.34] | 22.2 | 285 | 25.6 | 147 | 18.3 | 1.54 [1.21-1.97] | 0.893 | 0.865 |
| *Brain & nervous system* | | | | | | | | | | | | | | |
| Headaches | 43.2 | 117 | 53.7 | 109 | 39.5 | 1.80 [1.20-2.72] | 42.1 | 537 | 49.1 | 283 | 35.7 | 1.48 [1.20-1.83] | 0.674 | 0.094 |
| Concentration problems | 38.8 | 118 | 53.6 | 85 | 30.8 | 2.78 [1.84-4.23] | 39.7 | 554 | 50.3 | 218 | 27.5 | 2.54 [2.05-3.16] | 0.850 | 0.581 |
| Memory problems | 28.9 | 92 | 42.2 | 59 | 21.4 | 2.92 [1.86-4.57] | 29.8 | 430 | 39.1 | 150 | 18.9 | 2.50 [1.99-3.16] | 0.866 | 0.824 |
| General weakness | 22.9 | 71 | 32.4 | 49 | 17.9 | 2.47 [1.55-3.97] | 20.5 | 287 | 26.1 | 113 | 14.2 | 2.08 [1.61-2.71] | 0.300 | 0.115 |
| Dizziness | 20.8 | 62 | 28.6 | 47 | 17.0 | 2.00 [1.24-3.23] | 19.2 | 262 | 23.9 | 112 | 14.2 | 1.79 [1.38-2.32] | 0.542 | 0.154 |
| Balance problems | 17.6 | 51 | 23.5 | 41 | 14.9 | 2.15 [1.29-3.63] | 18.4 | 249 | 22.7 | 109 | 13.7 | 1.98 [1.51-2.60] | 0.996 | 0.635 |
| *Mental well-being* | | | | | | | | | | | | | | |
| Difficulty sleeping | 45.5 | 122 | 55.5 | 116 | 42.5 | 1.61 [1.08-2.41] | 45.9 | 564 | 51.3 | 329 | 41.9 | 1.32 [1.08-1.62] | 0.730 | 0.621 |
| Anxiety | 46.8 | 110 | 50.7 | 135 | 49.5 | 0.86 [0.56-1.31] | 45.8 | 565 | 51.5 | 326 | 41.5 | 1.26 [1.02-1.56] | 0.997 | 0.990 |
| Depression | 33.5 | 88 | 40.4 | 87 | 31.8 | 1.29 [0.84-1.99] | 33.8 | 417 | 37.9 | 241 | 30.5 | 1.19 [0.96-1.49] | 0.923 | 0.747 |
| *Ears, nose, and throat* | | | | | | | | | | | | | | |
| Congested nose | 34.8 | 97 | 42.7 | 85 | 29.9 | 1.78 [1.19-2.66] | 33.9 | 406 | 36.5 | 254 | 31.6 | 1.17 [0.95-1.45] | 0.378 | 0.474 |
| Ringing in ears | 17.2 | 49 | 21.7 | 41 | 14.5 | 1.90 [1.14-3.19] | 22.9 | 281 | 25.4 | 164 | 20.6 | 1.41 [1.04-1.67] | 0.711 | 0.298 |
| *Heart or circulation* | | | | | | | | | | | | | | |
| Irregular heartbeats | 18.9 | 63 | 28.3 | 36 | 12.9 | 2.72 [1.66-4.52] | 19.9 | 274 | 24.8 | 114 | 14.3 | 1.97 [1.52-2.57] | 0.753 | 0.151 |
| Leg pain when walking | 22.6 | 70 | 31.4 | 48 | 17.3 | 2.00 [1.25-3.24] | 16.7 | 233 | 21.1 | 93 | 11.7 | 2.12 [1.60-2.83] | 0.011 | 0.596 |
| *Eyes or vision* | | | | | | | | | | | | | | |
| Dry eyes | 22.6 | 61 | 27.4 | 57 | 20.4 | 1.83 [1.16-2.91] | 26.1 | 311 | 28.2 | 197 | 24.7 | 1.34 [1.07-1.69] | 0.996 | 0.712 |
| *Stomach or digestion* | | | | | | | | | | | | | | |
| Heartburn (reflux) | 21.6 | 62 | 28.4 | 51 | 18.3 | 2.25 [1.40-3.65] | 23.6 | 299 | 27.0 | 160 | 20.1 | 1.32 [1.05-1.68] | 0.981 | 0.718 |

[a]Hispanic/Latino with a positive Covid-19 test result versus Non-Hispanic/Latino with a positive Covid-19 test result.
[b]N Reported patient with symptoms.
[c]Adjusted P-Value calculation by Logistic Regression adjusted for age, sex, BMI, vaccine status, race/ethnicity, CCI, and time after COVID-19 test result.

**Table 7 Frequently reported symptoms by time between testing and survey receipt (n = 2539)**

| Symptomw | 3-9 months (n = 765) | | | | | 10-12 months (n = 1073) | | | | | More than 12 months (n = 701) | | | | |
|---|---|---|---|---|---|---|---|---|---|---|---|---|---|---|---|
| | COVID-19 test result | | | | | COVID-19 test result | | | | | COVID-19 test result | | | | |
| | Positive | | Negative | | Odds Ratio | Positive | | Negative | | Odds ratio | Positive | | Negative | | Odds ratio |
| | Na | % | Na | % | [95% CI] | Na | % | Na | % | [95% CI] | Na | % | Na | % | [95% CI] |
| *General symptoms* | | | | | | | | | | | | | | | |
| Fatigue/Tiredness | 247 | 58.7 | 129 | 38.4 | 2.29 [1.64-3.21] | 422 | 60.9 | 173 | 46.8 | 1.78 [1.34-2.35] | 159 | 56.2 | 159 | 38.4 | 2.25 [1.59-3.20] |
| Muscle & body aches | 167 | 39.7 | 83 | 24.9 | 2.11 [1.48-3.03] | 240 | 34.9 | 95 | 25.7 | 1.52 [1.12-2.07] | 106 | 37.5 | 102 | 24.6 | 1.82 [1.25-2.65] |
| Joint pains | 158 | 37.5 | 80 | 24.1 | 2.12 [1.48-3.04] | 238 | 34.6 | 85 | 23.0 | 1.83 [1.34-2.52] | 101 | 35.7 | 98 | 23.7 | 2.24 [1.52-3.32] |
| Shortness of breath | 130 | 30.8 | 56 | 16.7 | 2.14 [1.46-3.17] | 211 | 30.6 | 55 | 14.9 | 2.43 [1.72-3.46] | 87 | 31.0 | 47 | 11.4 | 4.16 [2.65-6.65] |
| Cough | 118 | 28.2 | 62 | 18.5 | 1.49 [1.01-2.21] | 170 | 24.8 | 66 | 17.9 | 1.57 [1.12-2.21] | 69 | 24.6 | 87 | 21.0 | 1.25 [0.82-1.89] |
| *Brain & nervous system* | | | | | | | | | | | | | | | |
| Headaches | 207 | 50.1 | 116 | 35.4 | 1.74 [1.24-2.44] | 328 | 49.2 | 150 | 41.6 | 1.32 [0.99-1.75] | 138 | 50.5 | 138 | 34.2 | 1.69 [1.17-2.44] |
| Concentration problems | 210 | 50.2 | 99 | 30.1 | 2.70 [1.92-3.83] | 343 | 51.2 | 107 | 29.6 | 2.49 [1.86-3.36] | 135 | 48.9 | 105 | 26.0 | 2.58 [1.79-3.74] |
| Memory problems | 173 | 41.6 | 69 | 20.9 | 2.98 [2.08-4.32] | 264 | 39.4 | 75 | 20.8 | 2.41 [1.76-3.33] | 97 | 35.3 | 71 | 17.6 | 2.26 [1.53-3.37] |
| General weakness | 122 | 29.3 | 54 | 16.5 | 2.06 [1.39-3.09] | 178 | 26.6 | 55 | 15.2 | 2.17 [1.53-3.14] | 69 | 25.1 | 61 | 15.1 | 1.83 [1.18-2.84] |
| Dizziness | 104 | 25.1 | 56 | 17.0 | 1.56 [1.04-2.34] | 165 | 24.8 | 53 | 14.7 | 1.90 [1.33-2.74] | 63 | 22.7 | 58 | 14.4 | 1.73 [1.12-2.68] |
| Balance problems | 99 | 24.1 | 55 | 16.8 | 1.89 [1.25-2.87] | 158 | 23.5 | 44 | 12.2 | 2.37 [1.61-3.54] | 47 | 17.0 | 58 | 14.4 | 1.57 [0.97-2.53] |
| *Mental well-being* | | | | | | | | | | | | | | | |
| Difficulty sleeping | 219 | 52.8 | 131 | 39.7 | 1.67 [1.20-2.32] | 342 | 51.0 | 158 | 44.3 | 1.21 [0.91-1.60] | 144 | 52.4 | 167 | 42.2 | 1.30 [0.91-1.84] |
| Anxiety | 216 | 51.9 | 138 | 41.9 | 1.55 [1.09-2.20] | 333 | 50.0 | 174 | 48.7 | 0.92 [0.69-1.24] | 142 | 51.8 | 162 | 40.8 | 1.26 [0.88-1.80] |
| Depression | 173 | 41.8 | 99 | 29.9 | 1.90 [1.34-2.72] | 253 | 37.7 | 122 | 34.0 | 1.07 [0.79-1.46] | 88 | 32.1 | 117 | 29.5 | 0.91 [0.62-1.34] |
| *Ears, nose, and throat* | | | | | | | | | | | | | | | |
| Congested nose | 180 | 43.0 | 95 | 28.4 | 1.81 [1.29-2.55] | 240 | 35.2 | 125 | 33.8 | 1.09 [0.82-1.45] | 91 | 32.4 | 127 | 30.9 | 1.02 [0.71-1.47] |
| Ringing in ears | 104 | 24.9 | 69 | 21.4 | 1.30 [0.89-1.92] | 163 | 24.1 | 72 | 19.6 | 1.23 [0.88-1.72] | 72 | 25.6 | 73 | 17.8 | 1.91 [1.27-2.90] |
| *Heart or circulation* | | | | | | | | | | | | | | | |
| Irregular heartbeats | 96 | 23.0 | 41 | 12.5 | 2.42 [1.57-3.89] | 172 | 25.5 | 51 | 13.9 | 1.95 [1.37-2.81] | 70 | 25.1 | 61 | 15.0 | 1.85 [1.21-2.85] |
| Leg pain when walking | 102 | 24.6 | 50 | 15.2 | 2.13 [1.41-3.27] | 153 | 22.6 | 45 | 12.3 | 2.11 [1.45-3.14] | 55 | 19.8 | 50 | 12.3 | 1.70 [1.05-2.77] |
| *Eyes or vision* | | | | | | | | | | | | | | | |
| Dry eyes | 123 | 29.6 | 80 | 24.2 | 1.63 [1.13-2.36] | 185 | 27.2 | 96 | 26.4 | 1.11 [0.81-1.53] | 74 | 26.9 | 85 | 20.8 | 1.72 [1.15-2.58] |
| *Stomach or digestion* | | | | | | | | | | | | | | | |
| Heartburn (reflux) | 120 | 28.9 | 62 | 18.7 | 1.82 [1.25-2.67] | 178 | 26.4 | 73 | 19.9 | 1.45 [1.04-2.02] | 73 | 26.4 | 84 | 20.7 | 1.17 [0.78-1.76] |

aResponder (Reported yes symptom); Odds ratio and 95% confidence interval by Logistic Regression adjusted for age, sex, BMI, vaccine status, race/ethnicity, CCI, and time after COVID-19 test, respectively.

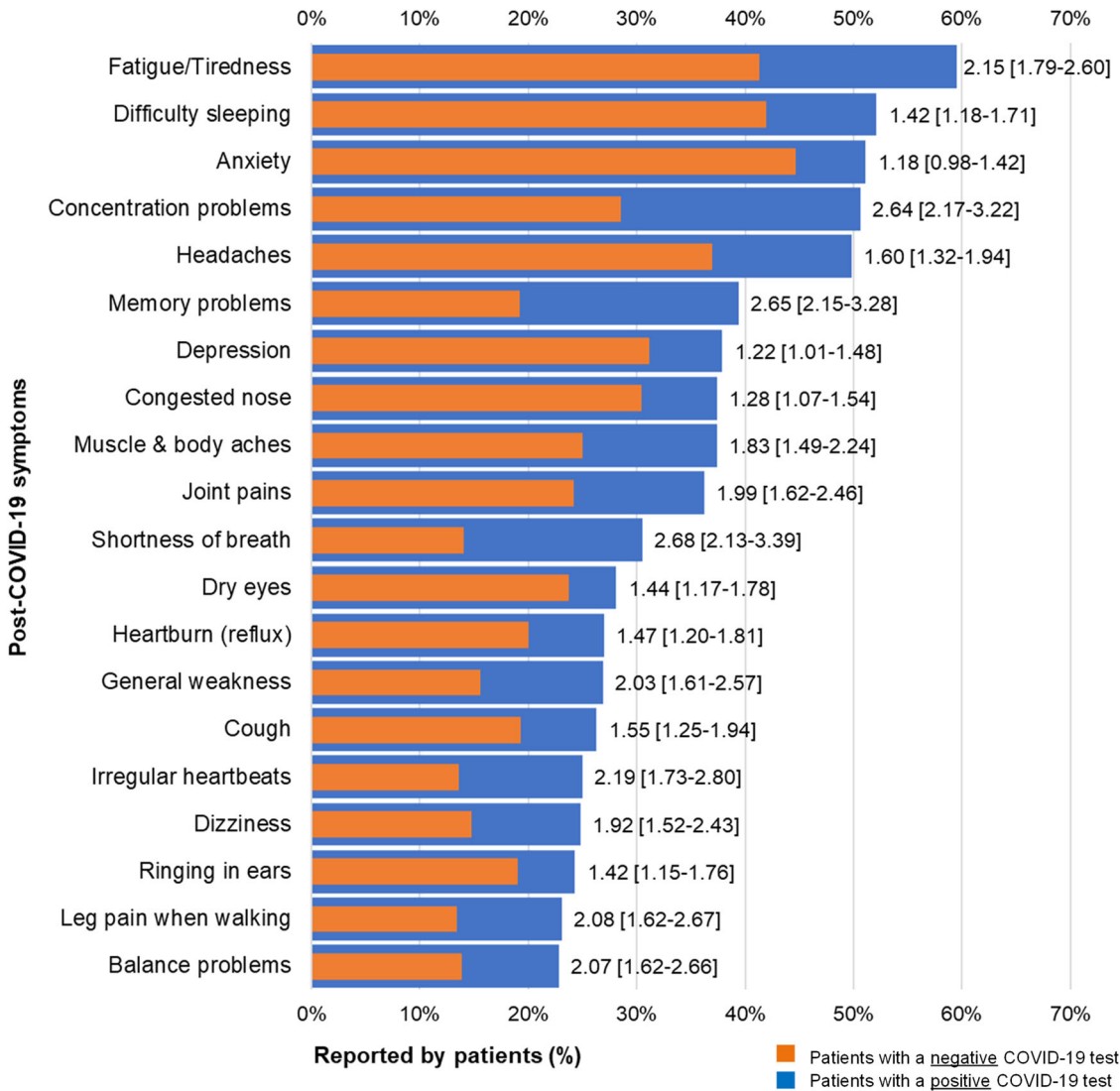

**Fig. 2 Frequently reported Post-COVID-19 symptoms comparing patients with and without COVID-19 (N = 2,539).** Figure 2 shows the top 20 most frequently experienced symptoms across the severity scale (mild, moderate, and severe). Bars in blue represent patients with a positive COVID-19 test and bars in red represent patients with a negative COVID-19 test. Adjusted odds ratios with [95% Confidence Interval] shown at the end of the bars suggest relationships between COVID-19 test result and reported symptom (see Table 2 for significance p < 0.05).

fatigue, and shortness of breath were higher than other symptoms, and COVID-19-positive patients were more than two times as likely to experience these symptoms compared to COVID-19-negative patients. Anxiety was the only symptom not significantly associated with a positive COVID-19 test result.

**Post COVID-19 conditions by age.** Table 3 (Supplementary Data 1) compares the prevalence of the twenty most common symptoms across three age groups, 18–34 (n = 810), 35-49 (n = 797) and ≥50 (n = 932). In all three age groups, the most common symptoms reported by the COVID-19 positive patients included fatigue (59.7% vs. 63.3% vs. 54.4%), concentration problems (51.8% vs. 56.1% vs. 43.1%), difficulty sleeping (54.5% vs. 54.8% vs. 45.8%). The COVID-19 positive patients in the age group 35–49 generally had significantly higher prevalence and ORs of these common symptoms than their counterparts in the other two age groups.

**Post-COVID-19 conditions by sex.** Regarding sex, the most common symptoms reported by both female (n = 1615) and male

(n = 924) COVID-19-positive patients included fatigue/tiredness (65.5% vs. 48.7%), headaches (57.7% vs. 36%), concentration problems (56.2% vs. 40.4%), memory problems (43.1% vs. 32.5%), difficulty sleeping (55.5% vs. 45.4%), and anxiety (57.7% vs. 39.3%). For both groups, COVID-19-positive patients had higher prevalence of each symptom than their COVID-19 negative counterparts. However, female COVID-19-positive patients had higher prevalence of each symptom than their male counterparts. In addition, the OR of each symptom was higher for females than for males, except for concentration problems (2.43 [1.94–3.05] vs. 3.08 [2.20–4.37]), difficulty sleeping (1.26 [1.01–1.57] vs. 1.62 [1.20–2.20]), and dry eyes (1.31 [1.03–1.66] vs. 1.65 [1.13–2.41]). Moreover, eight symptoms (fatigue/tiredness, joint pains, shortness of breath, concentration problems, memory problems, general weakness, irregular heartbeats, leg pain when walking) were more than twice as prevalent in female COVID-19-positive patients compared to their COVID-19-negative counterparts. In contrast, four conditions (fatigue/tiredness, shortness of breath, concentration problems, memory problems) were more than twice as prevalent in male COVID-19-positive patients compared to their COVID-19-negative counterparts (Table 4).

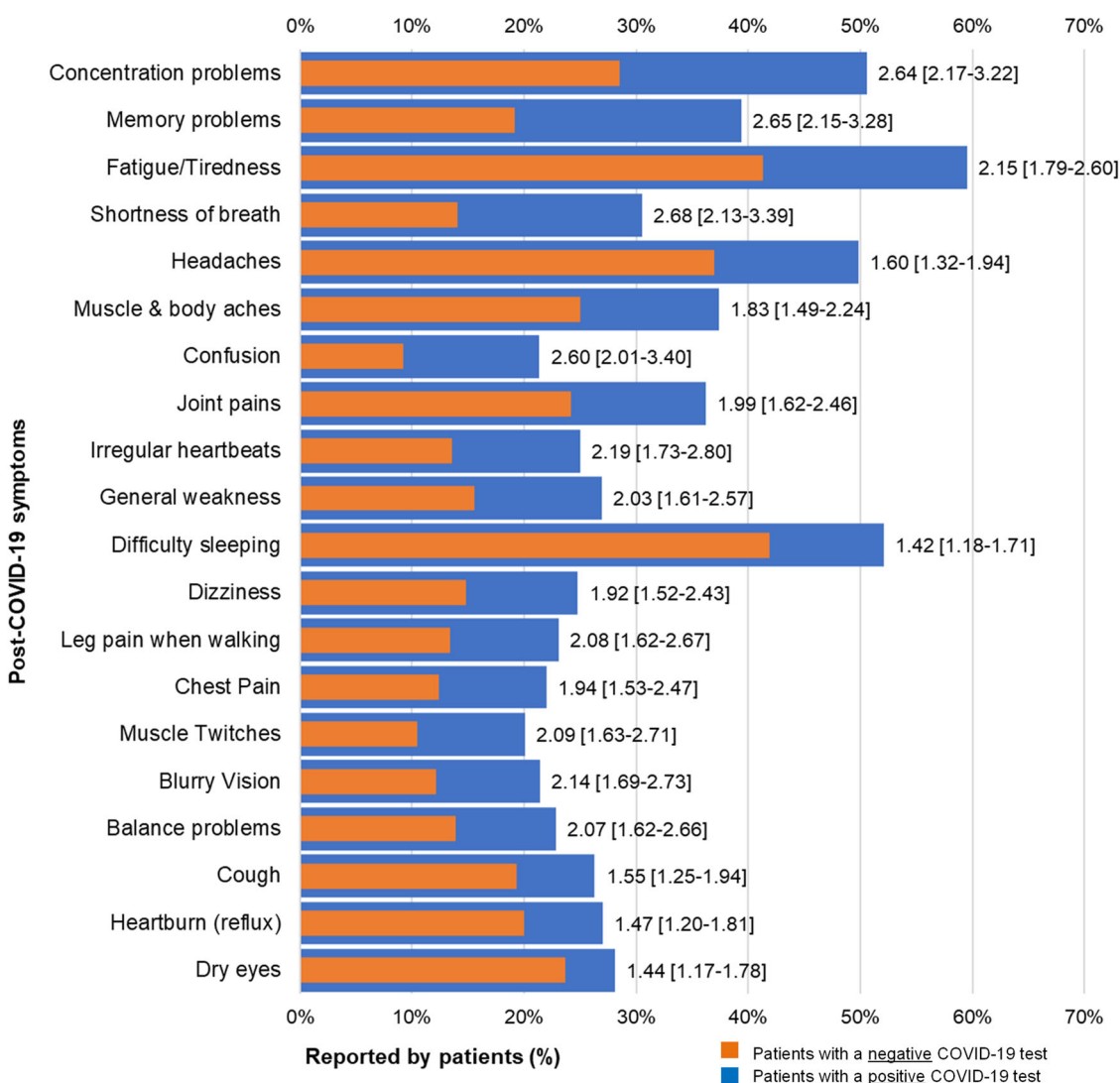

**Fig. 3** Comparing Post-COVID-19 symptoms with the largest differences between patients with and without COVID-19 ($N = 2539$). Figure 3 shows the top 20 largest differences in experienced symptoms across the severity scale (mild, moderate, and severe), in descending order by highest to lowest odds ratios. Bars in blue represent patients with a positive COVID-19 test and bars in red represent patients with a negative COVID-19 test. Adjusted odds ratios with [95% Confidence Interval] shown at the end of the bars suggest relationships between COVID-19 test results and reported symptoms.

**Post COVID-19 conditions by race.** In both groups (White and non-White), COVID-19-positive patients had a higher prevalence of each symptom. White COVID-19-positive patients had lower ORs of these common conditions than their non-White counterparts: fatigue (1.96 [1.61–2.38] vs. 3.12 [2.05–4.80]), headaches (1.49 [1.21–1.82] vs. 1.97 [1.29–3.05]), concentration problems (2.49 [2.03–3.07] vs. 3.13 [2.02–4.90]), difficulty sleeping (1.29 [1.06–1.57] vs. 1.74 [1.15–2.66]), and anxiety (1.11 [0.90–1.36] vs. 1.43 [0.92–2.21]). Moreover, among non-White patients, the ORs were greater than two for fourteen conditions (fatigue/tiredness, muscle & body aches, joint pains, shortness of breath, cough, concentration problems, shortness of breath, general weakness, dizziness, balance problems, irregular heartbeats, leg pain when walking, dry eyes, heartburn) as compared to three conditions (shortness of breath, concentration problems, shortness of breath) among White patients (Table 5).

**Post COVID-19 conditions by ethnicity.** In each group (Hispanic/Latino and non-Hispanic/Latino), COVID-19-positive patients had a higher prevalence of each symptom. Hispanic

COVID-19-positive patients had higher ORs of these common conditions than their non-Hispanic counterparts: fatigue/tiredness (2.28 [1.53–3.40] vs. 2.12 [1.73–2.60]), headaches (1.80 [1.20–2.72] vs. 1.48 [1.20–1.83]), concentration problems (2.78 [1.84–4.23] vs. 2.54 [2.05–3.16]), and difficulty sleeping (1.61 [1.08–2.41] vs. 1.32 [1.08–1.62]). Moreover, in the Hispanic group the ORs were greater than two for twelve conditions (fatigue/tiredness, muscle & body aches, joint pains, shortness of breath, concentration problems, shortness of breath, general weakness, dizziness, balance problems, irregular heartbeats, leg pain when walking, heartburn) as compared to six (fatigue/tiredness, shortness of breath, concentration problems, shortness of breath, general weakness, leg pain when walking) in the non-Hispanic group (Table 6).

**Post-COVID-19 conditions by time since COVID-19 test.** Patients were split into three groups based on when patients were tested for COVID-19: 3–9 months ($n = 765$), 10-12 months ($n = 1073$), and more than 12 months ($n = 701$). In each group, COVID-19-positive patients had a higher prevalence of each

symptom. The group at 3–9 months had the highest ORs of these common conditions: fatigue/tiredness (2.29 [1.64–3.21]), headaches (1.74 [1.24–2.44]) concentration problems (2.70 [1.92–3.83]), difficulty sleeping (1.67 [1.20–2.32]), and anxiety (1.55 [1.09–2.20]) (Table 7; Supplementary Data 2).

Supplementary Figure 3 shows the absolute differences in prevalence of top 10 most frequently experienced symptoms across the severity scale (mild, moderate, and severe) between COVID-positive and -negative patients at three time points. Supplementary Figure 3 also shows the adjusted odds ratios with 95% Confidence Intervals comparing the odds of each reported symptom between COVID positive and negative patients shown at the end of the bars. Symptoms were presented in descending order of the prevalence of the symptom for COVID-positive patients. Absolute prevalence differences were similar for fatigue (20.3%), concentration problems (20.1%), and memory problems (20.7%), and higher than absolute prevalence differences for other symptoms. Compared to COVID-negative patients, COVID-positive patients had higher prevalence and odds of each symptom at each time point. In addition, the prevalence of each symptom (except for concentration problems) reduced over time. Depression and congested nose were not significantly associated with COVID-test results past 9 months.

## Discussion

This study aimed to analyze the prevalence of PCC in non-hospitalized COVID-19 patients in primary care and compared the prevalence of PCC symptoms between patients with and without COVID-19. Our analysis revealed three major findings: First, post-COVID-19 conditions are very prevalent in this primary care population, independent of COVID-19. In particular, conditions impacting the brain and nervous system (e.g., concentration problems, headaches), mental health (e.g., anxiety, difficulty sleeping), and general well-being (e.g., fatigue/tiredness, shortness of breath) are common. Second, the burden of PCC is much higher among patients with COVID-19 compared to patients without COVID-19. Twenty common post-COVID-19 symptoms were prevalent in both COVID-19 positive and negative patients, and significantly more prevalent in patients with COVID-19, except for anxiety. Third, PCC was more prevalent in COVID-19-positive patients who were 35-49 years old, 3-9 months from their testing date, female, or from racial/ethnic minority groups than their COVID-19-negative counterparts by age, sex, time since test, and race/ethnicity.

In more detail, the prevalence of PCC varied with respect age, sex, time since COVID-19 test, race, and ethnicity. Concerning age, patients aged 35-49 years had a higher burden of PCC compared to younger and older patients. In general, the evidence on such disparity is inconclusive, although studies in hospitalized patients reported older age as a risk factor for developing PCC-related symptoms, while others point out that patients of all ages suffer long-lasting problems[28–31].

With respect to sex, female COVID-19-positive patients had a higher prevalence of each symptom than their male counterparts. Eight symptoms were more than twice as prevalent in female COVID-19-positive patients compared to their COVID-19-negative counterparts. By contrast, four symptoms were more than twice as prevalent in male COVID-19-positive patients compared to their COVID-19-negative counterparts. Our findings are in line with previous research that a higher proportion of female patients reported PCC-related symptoms than male patients[32,33]. However, since the time of writing, previous studies were much smaller, conducted outside the United States, and not exclusively focused on non-hospitalized patients in primary care. Future research should consider the role of immune response,

hormonal factors, and social or environmental factors in how PCC manifests among sexes.

With respect to race and ethnicity, COVID-19-positive patients had higher prevalence of each symptom in White and non-White patients and also in Hispanic/Latino and non-Hispanic/Latino patients. Twelve symptoms were more than twice as prevalent in Hispanic/Latino COVID-19-positive patients than their COVID-19-negative counterparts. In contrast, six symptoms were more than twice as prevalent in non-Hispanic/Latino COVID-19-positive patients compared to their COVID-19-negative counterparts. Similar findings were also observed in non-White and White patients, with more symptoms prevalent in the former group. Throughout the pandemic, disparities among non-white/Latino patients regarding exposure to SARS-CoV-2 and access to healthcare and social services were exacerbated[34–36]. Equitable access to quality primary healthcare services is critical; minority racial and ethnic groups generally have less access to care and were more likely to be exposed to COVID-19 (especially at the beginning of the pandemic) than White patients[37,38].

Concerning time since COVID-19 testing, COVID-19-positive patients 3-9 months out from their testing date had a higher prevalence of symptoms compared to patients with negative COVID-19 test results. Nine symptoms were more than twice as prevalent in the positive 3-9 month group, compared to seven in the positive 10-12 month group, and five in the positive 12+ month group. A reduction in the severity of some symptoms over time has been reported in other studies about PCC, but point out that neurological symptoms tend to persist[4,39]. In the present study, concentration problems and memory problems remain more than two times as likely to impact COVID-19-positive patients from 3 months of infection onwards.

More in general, most of the research on PCC has occurred among hospitalized patients, research regarding PCC among non-hospitalized patients in the United States primary care setting is emerging. As also shown in studies with hospitalized COVID-19 patients[28,40–45], the most prevalent symptoms among this non-hospitalized, primary care population were fatigue/tiredness, difficulty sleeping, and anxiety, followed closely by headaches and concentration problems. These symptoms were prevalent in at least half of COVID-19-positive patients. Existing studies with non-hospitalized patients were largely based on recruitment from social media groups[4,16,21], had smaller sample sizes[7,19,22,46–48], or lacked a COVID-19 negative comparison group[19,48,49]. Some research calls for the management of PCC in the primary care setting or describes the potential burden on healthcare systems[5,7], yet studies quantifying the burden of PCC symptoms in primary care have not been widely conducted. One study from a UK-based primary care database retrospectively analyzed PCC symptoms in a population-based cohort, and share some similar findings to the present study – PCC in non-hospitalized cohorts consists of heterogenous symptoms across the body, including fatigue[18]. Here, through our cross-sectional use of electronic health records and questionnaires, we show that many of the symptoms of long-COVID are prevalent in primary care, independent of COVID-19[50].

However, our study demonstrates that the great challenge facing primary care providers is to differentiate PCC from the acute sequela of COVID-19, previous comorbidities, preexisting conditions, as well as complications from prolonged illness, hospitalization, or isolation[51,52]. Evidence including the perspective of primary care physicians on how to best manage PCC has also been mounting and emphasizes the need for communication and trust between patient and provider to support management[53].

Based on the current evidence, disease management of PCC requires a holistic, longitudinal follow-up, multidisciplinary

rehabilitation services (e.g., family medicine, pulmonary, infectious disease, neurology), and the empowerment of affected patient groups[54]. Emotional support, ongoing monitoring, symptomatic treatment, and attention to comorbidities are cornerstones of this approach[55]. Primary care, and more specifically Family Medicine - with attributes like person-focused, comprehensive, and coordinated care—is theoretically very well-prepared to address those requirements[56–60]. Primary care clinicians know their patients, their lives, and their families and are in an optimal position to coordinate and personalize the treatment as well as the support needed. However, a comprehensive training program, including care pathways, guidance, and criteria to which patients should be referred, is necessary to support the primary care-led Post-COVID-19 response[61].

To the best of our knowledge, this is one of a few studies analyzing the prevalence of PCC in non-hospitalized COVID-19 primary care patients compared to primary care patients not diagnosed with COVID-19. Additionally, we included Spanish language preference questionnaires to encourage inclusivity for a variety of patients. Previous studies on PCC that include non-hospitalized patients experiencing symptoms for more than a year are uncommon in the literature. Methodological challenges were also common in prior work, including small study populations or exclusion of patients with negative test results.

One limitation of our study is a potential self-selection bias as patients with symptoms described as part of PCC are more likely to participate, evidenced by high rates of symptoms among patients with negative COVID-19 test results. Higher rates of symptoms in the 3-9 months since COVID-19 test group could also be attributed to self-selection bias since patients with perceived persistent symptoms may have been more inclined to participate in the questionnaire. Additionally, clinical data that would shed more light on the pathological mechanisms of PCC (e.g., chest x-rays, computed chest topography, inflammatory markers) were not included in this analysis. Finally, our questionnaire did not include questions about taste or smell disturbances, which have been cited frequently in the literature as common PCC symptoms[62,63]. We also did not include questions regarding the impact of symptoms on daily activities. Future research should include controls for symptoms that may have developed before the COVID-19 pandemic, include more clinical data, and develop methods for including older adults in outpatient studies.

## Conclusions
This study demonstrates that PCC is highly prevalent in non-hospitalized COVID-19 patients in primary care. However, it is important to note that PCC strongly overlaps with common health conditions seen in primary care, including fatigue, difficulty sleeping, and headaches. This makes the diagnosis of PCC in primary care even more challenging. There is an urgent need to strengthen the diagnosis and treatment of PCC in primary care.

## Competing interests
The authors declare no competing interests.

## Data availability
Based on the IRB protocol for data monitoring for this study, sharing data in an open-access repository does not meet the encryption standards for data sharing outside the University of Utah. Data will be made available to interested parties through an individual data transfer agreement upon request to the corresponding author. After the request, we will coordinate with the Technology Licensing Office to develop a data transfer agreement and ensure the transfer meets IRB protocol. The source data underlying Figs. 2 and 3 can be found in Table 3 and Supplementary Table 3, respectively.

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

## Acknowledgements

The research reported in this publication was supported in part by the National Center for Advancing Translational Sciences of the National Institutes of Health under Award Number UL1TR002538. This study was also supported Health Studies Fund, provided by the Department of Family and Preventive Medicine at the University of Utah. We would like to thank the participants of this study for taking the time to complete our questionnaire and share their experiences with our team. Additional thanks to external collaborators who offered their expertise on PCC clinical presentations, document translation, and questionnaire development. Finally, we would like to thank the colleagues who proof-read this manuscript prior to submission.

## Author contributions

B.K., K.S., J.L. and D.O. created the design and developed the first draft of the questionnaire. M.M. and C.S. supported the questionnaire development and delivery methodological (e.g., design) and administrative (e.g., redcap implementation). M.Z. provided data from the enterprise data warehouse (EDW). A.C., J. Wu and J. Wang carried out the data analyses and supported B.K., K.S., J.L., E.G. and D.O. with data interpretation. D.O. and E.G. have written the first draft of the manuscript with support from A.C. and J. Wang in the methods section. All authors have read the final manuscript and, overall, contributed extensively to this work.

## Competing interests

The authors declare no competing interests.
