## [Peer Review File · Communications Medicine]

Reviewers' comments:

Reviewer #1 (Remarks to the Author):

Ose et al, studied post covid symptoms/conditions (PCC) in a group of non-hospitalized patients visiting clinic for symptoms and had a COVID-19 test done and compared patients who had a positive and a negative results. PCC was assessed based on a questionnaire delivered to all study participants. The authors found that, despite that PCC is found both in patients with positive and negative COVID-19 tests, the individual and combined prevalence of PCC was significantly higher among patients who had a positive COVID-19 test. The prevalence, odds ratios, and associated symptoms of PCC vary when stratified by age, sex, and race/ethnicity. Patients with positive COVID-19 test results who are 35-49 years old, female, or from racial/ethnic minority groups report symptoms more often and have higher odds ratios than their respective comparison groups.

The authors concluded that PCC in its current definition poses a significant overlap with other healthcare conditions commonly seen in the outpatient setting which increase the difficulty of identifying the condition in common daily clinical practice, and suggest that, to manage PCC, a holistic, longitudinal follow-up, multidisciplinary rehabilitation services are required. Primary care, particularly family medicine, is well-prepared to address these requirements but needs a comprehensive training program to support the primary care-led response.

The study's strengths include its large sample size and its longitudinal design. The findings highlight the significant burden of post-COVID-19 symptoms in non-hospitalized patients, with fatigue being the most commonly reported symptom. The study also identified certain demographic factors, such as female gender and non-white race that were associated with higher symptom severity and longer duration. Another strength is that it has a control group and that was studied for a long period.

These are my comments

Comments to the authors:

1. One potential criticism of this discussion is that it focuses primarily on the prevalence and demographics of post-COVID-19 conditions, rather than discussing into the underlying causes and mechanisms of these conditions. While it is certainly important to understand who is most affected by these conditions and what symptoms are most common, it is equally important to understand why these conditions are occurring and what can be done to treat and prevent them because, as for now, the management of this condition is not standardized.
2. Additionally, the discussion could benefit from more specific examples or case studies of how primary care clinicians are addressing post-COVID-19 conditions and what has been effective in managing them.
3. The authors developed a hypothesis based on COVID-19 test results only, however, PCC is more often reported in patients who had a clinically evident COVID-19 especially those hospitalized or had serious disease course. In the current study, however, the hypothesis face two major challenges, first, none of the patients were hospitalized and it is not clear what the clinical course after the test result was (symptoms/treatment/duration of symptoms and length of therapy). Second, most of the patients come from an earlier phase of the pandemic during which the test kits used were more prone to false positive as well as false negative results. As such this may contribute to the overlap seen and is considered a major limitation to the current study protocol
4. It is important to emphasize that the differentiation between post-COVID-19 symptoms and other underlying medical conditions can be a challenging task for healthcare professionals. Authors should spend some effort discussing the differences between patients with and without PCC regarding not only the results of a COVID-19 test results but also their Comprehensive medical history (pre-existing conditions or recent illnesses), physical examination, prior diagnostic tests (blood tests, imaging studies) and Pattern of symptoms.
5. As suggested by the authors, treatment of post-COVID-19 conditions may require a multidisciplinary approach, however I suggest also highlighting the role of infectious diseases,

neurology, and pulmonary.

6. Do the authors have any data that can shed more light on pathological description of the COVID-19 positive cohort, such as chest X-rays, computed topography of the chest, and inflammatory markers? While I understand the difficulty of collecting such information considering the study design, it is important to highlight that in the limitations section.

7. In the study, it is described that, overall, women were more affected by PCC than men. Personally, I believe that a more comprehensive explanation to this finding could be plausible and beneficial. There are some studies, which attempt to explain why more women experienced post-COVID-19 symptoms compared to men suggesting that the differences may be explained by differences in immune response, hormonal factors, and social and environmental factors. Women generally have stronger immune responses than men, which may contribute to increased incidence of autoimmune disorders and post-viral syndromes. Additionally, hormonal factors such as estrogen levels may play a role in the development and severity of post-COVID-19 symptoms. Social and environmental factors, such as greater caregiving responsibilities and increased exposure to the virus in certain occupations, may also be contributing factors. However, more research is needed to fully understand the sex differences in post-COVID-19 symptom.

8. Do the authors have data regarding the prevalence of autoimmune diseases in male and female in the cohort? There have been reports of individuals with pre-existing autoimmune conditions experiencing more severe symptoms and a longer duration of illness following a COVID-19 infection. Additionally, some of the symptoms of post-COVID-19 syndrome, such as fatigue, joint pain, and brain fog, overlap with symptoms of autoimmune diseases.

Reviewer #2 (Remarks to the Author):

This study that aimed to analyze the prevalence Post-COVID-19 conditions in non-hospitalized COVID-19 primary care patients and compared it with primary care patients not diagnosed with COVID-19 is well conducted by the author. However, there are minor corrections required. See below:

Methodology: Line 110, delete the second 'utilizing' repetition

Results: Line 186, delete this statement 'concentration problems (50.6% vs. 28.5%, 2.64 [2.17-187 3.22])....' Repetition.

Line 197 delete the word 'prevalence'

Line 225, the subtitle 'Post-COVID-19 conditions by time since COVID-19 test' was not discussed in the discussion section of the manuscript. The 3-9 months with the prevalence of symptoms may suggest that the more recent the infection, the higher the prevalence of PCC symptoms.

Discussion: Line 239, delete the words '..with a..'

Line 249-252, I will suggest supporting this statement 'The great challenge is to differentiate Post-COVID-19 conditions from the acute sequela of COVID-19, previous comorbidities, understanding preexisting conditions, as well as complications of prolonged illness, hospitalization, or isolation' with the reference below:

Sanyaolu A, Marinkovic A, Prakash S, Zhao A, Balendra V, Haider N, Jain I, Simic T, Okorie C. Post-acute sequelae in COVID-19 survivors: an overview. SN comprehensive clinical medicine. 2022 Apr 6;4(1):91.

Line 265, rewrite '..this inconclusive..' as '..this is inconclusive..'

Line 272, this statement may not be correct '..However, previous studies were much smaller...' because I found a previous large study that has been done on both hospitalized and non-hospitalized

cohort groups. See this reference below:

Rohrer-Meck K, Marchena D, Rubin-Miller L, Bregman H, Lo J, et al. Nearly 1 in 10 COVID patients seek treatment for long-term symptoms. *Epic Health Res Netw*. 2021. <https://ehrn.org/articles/nearly-1-in-10-covid-patients-seek-treatment-for-long-term-symptoms>. Accessed 3 Dec 2021.

The study also reported females have a higher proportion of symptoms than males. So this study is in line with the result obtained here. I will suggest citing this reference to support the statements.

Strength and Limitations: In line 284, remove the word 'systematically'

Reviewer #3 (Remarks to the Author):

The role of primary care clinicians will be vital to ensuring that patients with Post-COVID Conditions receive the care they need. While this paper highlights the many symptoms patients are experiencing, it would be strengthened by linking the patient report of symptoms to clinician visits. Are the authors able to determine if patients reporting these symptoms are also seeking medical care for the symptoms? If there is not evidence of medical care for the symptoms, this might also indicate an unmet need for patients. Previous studies in this area tend to focus on either evidence of Post-COVID conditions in health records, or on patient reported outcomes, few studies are able to do both. Refocusing the paper to link the clinical care and patient reported symptoms would add to a gap in the literature.

There are a number of other studies that have reported post-COVID conditions among non-hospitalized patients, including from community based surveys. It is recommended that the authors update the references and include this in the background and discussion.

The unique aspect of this survey is that COVID-19 patients are identified from primary care clinics indicating that they had milder COVID-19 illness.

Please include details on when (dates) the survey was conducted.

Is there any information on why patients were tested for COVID-19? For the COVID-negative group, were they tested because of presenting with symptoms or part of a general screening program of all patients being seen in the clinical setting?

A general comment is that it is often not clear who the comparison group(s) is in the reporting of the results or the conclusions. An example of this is in reporting of results by race. The authors report the observed differences among COVID + and COVID - patients who are white and make the statement that the OR are lower in this group than in the non-white group. However, the results among the non-white group are also comparing COVID+ and COVID -, not comparing the prevalence of symptoms in COVID + patients who are white to COVID + patients who are non-white. In table 5, there is no significant difference in reported symptoms among the COVID + patients by race. These results are not clearly communicated.

Reporting the absolute and relative differences between the COVID-positive and COVID-negative groups is appropriate. However, the authors have not reported this clearly for the reader. Figure 2 and 3 are difficult to interpret. Recommend revising these figures to only report the absolute difference, anchored on zero, for each condition and not include the prevalence among the COVID + or the odds ratio. The absolute difference needs to include 95% CI.

For all estimates, include 95% CI of SE.

The authors compare differences by sex and race and ethnic group but not age or time since testing. Recommend adding age and time since testing comparisons.
Please add dates, location, study to the title of each table and figure.
Please add number of missing and sample sizes for groups in footnotes
Are all the reported OR adjusted OR? If so, please state this and clarify what is adjusted for in the models.

Recommend adjusting for BMI, vaccination status, CCI.
Did the survey include any measures of duration of symptoms or impact on daily activities? If not, recommend adding this to the limitations.

As the authors note, many of these symptoms are common and not unique to SARS-CoV-2 infection. A discussion of the role of primary care clinicians and their role in caring for patients with these symptoms in general would strengthen the paper. Is the main message that because of COVID-19, more patients may be experiencing these symptoms and therefore seeking medical care and that primary care clinicians may have an increase in patient volume?

The authors conclude with an urgent need to diagnose and care for Post-COVID conditions in the primary care setting, but the results presented do not offer any guidance in this area beside indicating that patients maybe experiencing symptoms

Response to Reviewer comments

Reviewer 1:

Reviewer Comment	Author Response
1. Discussion focuses primarily on prevalence and demographics of Post Covid Conditions (PCC), rather than discussing underlying causes	The authors agree that discussing the underlying causes and mechanisms of this condition are important, and that the lack of options for management of this illness is concerning. However, this was a study about the prevalence of symptoms in the early covid-19 era and seeks to describe symptoms of PCC in the context of primary care. In the discussion, we describe options for improving management in primary care, supported by other literature, particularly through training and the creation of standard guidelines for care. As far as we know, there is not consensus on why these conditions occur or what can be done to treat them.
2. Discussion could benefit from more specific examples and case studies of how PCPs are addressing PCC and what has been effective in managing them	In the discussion, citations 38-42 describe the role of family medicine in managing post covid conditions and what has been done thus far to manage patients with PCC.
3. hypothesis challenges: (1) no hospitalized patients (2) patients come from early in the pandemic when testing was less robust	(1) The authors agree that PCC is more prevalence in hospitalized cohorts and point out that most of the research about PCC comes from this groups in the introduction. However, the aim of the study was to consider non-hospitalized patients to better understand the prevalence of PCC in non-hospitalized groups, because of a lack of representation in the literature. The clinical course after the test result is captured by the survey responses about symptoms, they experienced for the following 3 months. We did not ask about treatments or therapies in this paper because the focus is on prevalence of symptoms. (2) Patients coming from an earlier phase in the pandemic when testing was not as sensitive or specific has been added as a limitation of the study. The test results included in the study were all PCR tests, as stated in the methods section (inclusion/exclusion criteria).
4. (1) Emphasize the differentiation between PCC symptoms and their medical conditions is challenging (2) Discuss the differences between patients with and without PCC by test result but also by comprehensive medical history	(1) The major finding of this paper is that differentiation between PCC and other medical conditions is challenging, we agree with this point from the reviewer. We have strengthened the background, discussion, and conclusion by emphasizing this point.

	(2) Participants were asked about symptoms that developed/remained from initial covid-19 infection that were not present before their infection. Because this is also a population level study based on electronic health records, it was not possible to have a very comprehensive medical history of each patient. However, we controlled for pre-existing conditions (as well as other factors like age, sex, etc.) at the population level using the Charlson Comorbidity Index (CCI), which is based on information in the EHR.
5. Highlight the role of infectious diseases, neurology, pulmonary in treatment of PCC	We included a highlight of these fields in the discussion of multidisciplinary rehabilitation
6. Shed more light on pathological description of COVID-19 positive cohort	Added a statement on clinical limitations to the strengths and limitations section as an area for future study.
7. More research is needed to fully understand the sex differences in PCC	The authors agree that more research should be conducted regarding sex differences and how PCC manifests, and a statement about direction of future research has been added to the discussion.
8. More information about the prevalence of autoimmune disorders in male/female cohort, some of the symptoms overlap	The major finding of our paper is that the overlap of PCC symptoms with other concerns makes identifying patients with PCC a challenge, this point regarding autoimmune disease overlapping with PCC strengthens this point. We added a statement on clinical limitations to our strengths and limitations section to underscore the need for future research inclusive of more clinical data.

Reviewer 2:

Reviewer Comment	Author Response
1. Methods: line 110 remove "utilizing" repetition	Repetition removed
2. Results: line 186 delete repetition	Repetition removed
3. Results: line 193 delete prevalence	Repetition removed
4. Discussion: Line 225 post covid conditions by time since Covid-19 test was not discussed	Added time since test into summary of results in discussion
5. Discussion: line 239 delete "with a"	Done
6. Discussion: lines 249-252 add Sanyaolu reference	Reference included
7. Discussion: line 265 rewrite for clarity	Done
8. Discussion: line 272, check validity of claim with provided reference	The reference provided includes both hospitalized and non-hospitalized patients, but we chose not to include it because they used a different case definition for long haul

	COVID-19 than the authors of this study. Since the time of writing, one larger study based exclusively on non-hospitalized patients has been published based on UK data (cited in background). This study remains the largest population based study of an exclusively non-hospitalized/primary care cohort in the United States as far as the authors are aware.
9. Discussion: add reference about female prevalence	We chose not to include this reference since the authors used a different definition of long covid than the authors of the present study. The observation that female sex is associated with long covid is supported by 3 additional citations.
10. Strengths and limitations: line 284 remove “systematically”	Removed

Reviewer 3:

1. Are authors able to determine if patients reporting the symptoms are also seeking medical care for the symptoms? This would be an unmet need to explore	We did not collect data on if patients were seeking care for their reported symptoms. This has been added as an area for future research.
2. Update references about non-hospitalized patients	To the best of our knowledge, one larger study based on exclusively non-hospitalized patients using primary care data has been published since the time of writing. However, this study was conducted in the UK. This citation is included in the background.
3. (1) include details on when the survey was conducted (2) info on why patients tested for COVID 19	(1) Both the English and Spanish versions of the survey started on 8/31/2021 and ended 11/15/2021. Accordingly, we added this information on page 5. (2) Participants reported on symptoms they experienced in the last week, so participants tested for COVID-19 because they were symptomatic – described in methods.
4. “A general comment is that it is often not clear who the comparison group(s) is in the reporting of the results or the conclusions. An example of this is in reporting of results by race. The authors report the observed differences among COVID + and COVID – patients who are white and make the statement that the OR are lower in this group than in the non-white group. However, the results among the non-white group are also comparing COVID+ and COVID -, not comparing the prevalence of symptoms in COVID + patients who are white to COVID +	The adjusted p values in Table 5 were calculated to compare white and non-white COVID-positive patients using logistic regression after the adjustment of age, sex, BMI, vaccine status, ethnicity, CCI, and time after COVID-19 test, as done in a similar fashion to supplemental Table 3B for age, Table 4 for sex, Table 6 for ethnicity, and supplemental Table 7B for time. Accordingly, we have made a correction in the footnote for clarification. We apologize for the confusion.

patients who are non-white. In table 5, there is no significant difference in reported symptoms among the COVID + patients by race. These results are not clearly communicated.” Clarify comparison groups in the reporting of results and clarify table 5 (race) significance	
5. “Reporting the absolute and relative differences between the COVID-positive and COVID-negative groups is appropriate. However, the authors have not reported this clearly for the reader. Figure 2 and 3 are difficult to interpret. Recommend revising these figures to only report the absolute difference, anchored on zero, for each condition and not include the prevalence among the COVID + or the odds ratio. The absolute difference needs to include 95% CI.” clarify figure 2 and 3 – recommend only including abs difference and CI	We realized there was a lack of clear interpretations of eFigures 2 & 3. In fact, we included absolute prevalence differences in Figure 3, rather than in eFigure 2, as we previously stated. eFigure 2 presented odds ratios given in Table 2 to show the group difference in odds of each symptom. We apologize for the confusion. Accordingly, we added detailed interpretations in the Results section on page 11. We also revised the format of eFigure 2 for clarity.
6. for all estimates, include 95% CI of SE	Thank you for the suggestion. Accordingly, we have provided confidence intervals of all continuous variables (age, CCI, and BMI) in Table 1.
7. authors compare differences by sex and race/ethnicity but not age, or time since testing – add age and time since testing comparisons	Thanks for your comment. That is correct that we did not provide adjusted p values in Table 3 and Table 7 as we did in tables 4-6, primarily due to the table space limit. Accordingly, we have included the adjusted p values in both supplemental tables 3B and 7B.
8. recommend adding dates, location, study to the title of each table and figure	Thank you for the suggestion. The dates, location, and study details were described in the manuscript with reference to the tables and figures as applicable.
9. please add number of missing and sample sizes for groups in the footnotes	Thanks for your suggestion. We have added this information in the footnote of Table 1. Specifically, CCI had n=4 missing values (2 in the COVID-positive group, 2 in the COVID-negative group). Smoking status had n=16 missing values in the COVID-positive group and n=9 missing values in the COVID-negative group. BMI had n=159 missing values in the COVID positive group and n=89 missing values in the COVID negative group.
10. clarify if all the reported OR are adjusted OR and state what is adjusted for in the models	Yes, we made sure that all reported ORs were adjusted ones, as we stated in the Statistical analysis Section on page 8 as “Logistic regression was conducted to assess the association of Post-COVID-19 conditions with COVID-19, after the adjustment of

	aforementioned covariates” as well as in the footnote of tables 3-7.
11. Recommendation to adjust for BMI, vax, CCI	Thanks for your comment. Sex and CCI were already adjusted for in all our analyses Please see the footnotes below tables 4-6. We double checked our codes for logistic regression and made sure that we included BMI, as stated in the Statistical analysis Section on page 7 as “.... COVID-19 test result, Charlson comorbidity index (CCI), and body mass index)”. We realized that we failed to include BMI in the footnote of tables 3-7. Accordingly, we added this information.
12. “Did the survey include any measures of duration of symptoms or impact on daily activities? If not, recommend adding this to the limitations.” (1) clarify measures of duration of symptoms or (2) impact on daily activities	1) Our measure of duration of symptoms is recorded in the time since covid test table. (2) We did not ask about impacts on daily activities for this analysis. Added to limitations
13. Discussion on the role of primary care clinicians and their role in caring for patients with symptoms	We discussed the role of primary care in the discussion – a place where patients are already well understood by providers and providers are concerned with the whole person – but explain that more training will be needed for them to successfully address this issue.
14. Results do not offer guidance on diagnosing and caring for PCC patients	We discussed what diagnosing and caring for patients with PCC might look like in the discussion section (holistic care, follow-up, multidisciplinary action, etc.).

REVIEWERS' COMMENTS:

Reviewer #1 (Remarks to the Author):

The authors addressed our concerns. This report is an important contribution to our current understanding of Covid 19.

Reviewer #2 (Remarks to the Author):

The authors have made my corrections.

Reviewer #3 (Remarks to the Author):

Thank you for making these revisions. No additional comments.